# Epidemiological Profile of Hospitalized Patients with Cystic Fibrosis in Brazil Due to Severe Acute Respiratory Infection during the COVID-19 Pandemic and a Systematic Review of Worldwide COVID-19 in Those with Cystic Fibrosis

**DOI:** 10.3390/healthcare11131936

**Published:** 2023-07-04

**Authors:** Leonardo Souza Marques, Matheus Negri Boschiero, Nathália Mariana Santos Sansone, Letícia Rulli Brienze, Fernando Augusto Lima Marson

**Affiliations:** Laboratory of Molecular Biology and Genetics, São Francisco University, Bragança Paulista 12916-900, SP, Brazil; leonardo.marques@mail.usf.edu.br (L.S.M.); boschiero.matheus@gmail.com (M.N.B.); nathaliasansone@hotmail.com (N.M.S.S.); leebrienze@outlook.com (L.R.B.)

**Keywords:** Brazil, CFTR, epidemiology, p.Phe508del, SARS-CoV-2, mucoviscidosis, severity, systematic review, worldwide

## Abstract

Since the onset of the coronavirus disease, COVID-19 pandemic, concern arose for those who might be at higher risk of a worse COVID-19 prognosis, such as those with cystic fibrosis (CF). In this context, we evaluated the features of hospitalized patients with CF due to severe acute respiratory infection (SARI) in Brazil and we also performed a systematic review including all the studies published from the beginning of the first case of COVID-19 (17 November 2019) to the date of this search (23 May 2022) which included, concomitantly, patients with CF and COVID-19 in the worldwide population. In our Brazilian data, we evaluated the period from December 2019 to March 2022, and we included 33 demographical and clinical patients’ features. We classified the patients into groups: (G1) SARI due to another viral infection than severe acute respiratory syndrome coronavirus 2 (SARS-CoV-2) (23; 5.4%), (G2) SARI due to an unknown etiological agent (286; 67.1%), and (G3) SARI due to SARS-CoV-2 infection (117; 27.5%). The individuals in G3 tended to be older, especially over 50 years old, and presented a higher prevalence of dyspnea, peripheral capillary oxygen saturation (SpO_2_) <95%, and cardiopathy. The highest prevalence for intensive care unit (ICU) treatment (52; 44.4%) and invasive mechanical ventilation (29; 24.8%) was for patients in G3. Almost half of the patients in G3 died (51; 43.6%); in contrast, none in G1 died. However, we observed 43 (15.0%) deaths in G2. In addition, 12 (4.2%) and one (0.9%) death not associated with SARI occurred, respectively, in the G2 and G3. The patients who died due to SARS-CoV-2 infection had a higher frequency of SpO_2_ <95% (46; 90.2%), ICU treatment (34; 66.7%), and invasive mechanical ventilation (27; 52.9%) when compared to those who recovered. The systematic review comprised a total of 31 papers published as observational studies. These studies comprised 661,386 patients in total, including children, adults, and elderly age groups. However, only 19,150 (2.9%) patients were diagnosed with CF and, from these patients, 2523 (0.4%) were diagnosed with both CF and COVID-19. It was observed that the most common outcome was the need for hospitalization (n = 322 patients with CF), and the need for oxygen support (n = 139 patients with CF). One hundred patients with CF needed intensive care units, fifty patients needed non-invasive mechanical ventilation support, and only three patients were described as receiving invasive mechanical ventilation support. Deaths were described in 38 patients with CF. Importantly, lung-transplanted patients with CF represented an increased risk of death in one publication; in accordance, another study described that lung transplantation and moderate to severe lung disease were independent risk factors for severe outcomes after SARS-CoV-2 infection. In contrast with the literature, in conclusion, Brazilian patients in G3 presented a severe phenotype, even though most of the other studies did not observe worse outcomes in patients with CF and COVID-19.

## 1. Introduction

Cystic fibrosis (CF; OMIM: no. 219700), one of the most common Mendelian autosomal recessive disorders, is characterized by two aberrant copies of the Cystic Fibrosis Transmembrane Conductance Regulator (*CFTR*; region 7q31.2) gene [1], which ultimately alter the CFTR protein, an ion channel responsible for transporting chloride and bicarbonate across epithelial surfaces, leading to the dysregulation of endothelial cells and decreased pH in airway surface liquid [1,2,3]. Most CF symptoms are related to the respiratory tract, such as mucus accumulation and difficulty cleaning the respiratory tract, which could lead to an enhanced risk of chronic infections, inflammation [2,3], and death [4].

Since the rise of the coronavirus disease, COVID-19 pandemic, concern arose for those who might be at higher risk of worse COVID-19 prognosis, such as those with underlying conditions, which included Down syndrome, older age, systemic arterial hypertension, and diabetes mellitus [5,6,7] or even neglected populations, such as Indigenous peoples and Black individuals [8,9,10,11]. Nevertheless, the impact of COVID-19 on individuals with CF is not fully understood yet. Previous studies observed that viral infections might lead to pulmonary exacerbations, and, in the 2009 H1N1 pandemic, patients with CF presented higher case fatality rates when compared with the general population [3,12,13,14,15,16,17]. Although it is plausible that patients with CF might bear an increased burden of COVID-19, more observational studies are necessary to evaluate the outcomes of the severe acute respiratory syndrome coronavirus 2 (SARS-CoV-2) infection among those with CF in the real world.

Recent studies tried to evaluate how SARS-CoV-2 affected individuals with CF [18,19,20]. However, in the literature, there are some contrasting results; for instance, Bain et al.’s study (2021) comprised a total of 105 children with COVID-19 and CF, and most of them (71.0%) were managed in the community with mild COVID-19, with no need for hospitalization [19]. On the other hand, Naehrlich et al.’s study (2021) comprised 130 patients with COVID-19 and CF, and the incidence of COVID-19 was higher in the age groups <15, 15–24, and 25–49 years old when compared to the general population [18]. In addition, individuals with CF had a higher hospitalization rate than the general population [18]. Finally, a multicenter study that took place in Italy comprised only 30 patients, 16 with CF and COVID-19 and 14 with only CF; however, the authors did not find any difference in hospitalization rate, need for intensive care unit (ICU) admission, or even forced expiratory volume in the first second between the two groups of patients [20].

Only two studies comprised Brazilian individuals with CF and COVID-19. Bain et al. (2021) described a population from 13 countries, such as Argentina, Germany, France, and Brazil; however, the cases were not specified regarding their country of origin in order to maintain confidentiality [19]. In their study, Bain et al. (2021) described the features of 105 patients with CF and COVID-19 aged less than 18 years; the authors described how a major proportion of the patients were treated in the community, and only 24 patients were hospitalized. Also, one death occurred and was not associated with COVID-19 [19]. In another study, the authors described SARS-CoV-2 infection in a Brazilian post-transplanted patient with CF who had a concurrent infection with an adenovirus [21]; the patient presented clinical recovery after supportive treatment without directed management for COVID-19.

Thus, our primary aim was to perform an observational study in Brazil to describe the demographical and clinical characteristics of hospitalized individuals with CF affected by severe acute respiratory infection (SARI) due to another known respiratory virus than SARS-CoV-2, due to an unknown etiological agent, and due to SARS-CoV-2 (COVID-19). Also, our secondary aims were (i) to associate the demographical and clinical characteristics of hospitalized individuals with CF affected by SARI in the group of patients with COVID-19 with the outcomes (death or clinical recovery), (ii) to characterize the virus profiles from the patients with CF classified as SARI due to another known respiratory virus, as well as the co-detection in the patients with positive SARS-CoV-2 infection, and (iii) to perform a systematic review of papers regarding individuals with COVID-19 and CF (worldwide population) to evaluate the amount of evidence there is in the literature regarding these two diseases in the first two years of the COVID-19 pandemic in Brazil. 

## 2. Materials and Methods

### 2.1. Systematic Review of COVID-19 among Those with CF—Worldwide Population

The systematic review aimed to highlight the clinical progression, short-term impact, and other peculiarities of SARS-CoV-2 infection in people with CF (worldwide population). Initially, we conducted an advanced search in the PubMed virtual library with the following words (descriptors): ((COVID-19 OR COVID OR Coronavirus Disease OR Coronavirus Disease 2019 OR SARS-CoV OR SARS-CoV-2 OR SARS OR severe acute respiratory syndrome) AND (Cystic Fibrosis OR CFTR OR Cystic Fibrosis Transmembrane Regulator)) to search for English-language publications about the subject, generating the search data presented as Appendix A.

The publication date of the papers was defined from the beginning of the first case of COVID-19 (17 November 2019) to the date of this search (23 May 2022). Only studies published using English, including patients with CF and COVID-19, were included in the data collection. At the end of the search, 422 publications on the subject were obtained; after reading the title and the full abstract, 391 papers were excluded for not meeting the inclusion criteria. Therefore, in this study, we included thirty-one (N = 31) studies that comprised the inclusion criteria. The selection of papers was made by three authors (LSM, MNB, and FALM) blindly using the tool Rayyan (https://www.rayyan.ai/ (accessed on 19 June 2023)). All three authors agreed with the inclusion of a paper for it to be included in our study; if any disagreement was observed, the paper was discussed between the authors to confirm the inclusion or exclusion. The information regarding authorship, journal of publication, aim, sample (population evaluated in the study), study type, results, methods, and conclusion of the studies included in the systematic review were collected and summarized. We followed the PRISMA (Preferred Reporting Items for Systematic Reviews and Meta-Analyses) guideline for systematic reviews [22].

### 2.2. Epidemiological Study of the Brazilian Hospitalized Patients with CF and SARI

We performed a cross-sectional and analytical epidemiological analysis using Brazilian epidemiological (demographical and clinical) data available at OpenDataSUS (https://opendatasus.saude.gov.br/ (accessed on 23 March 2022)). The Brazilian Ministry of Health computed the data according to the surveillance data of SARI and from the platform of the Information System for Epidemiological Surveillance of Influenza (SIVEP-Gripe; in Portuguese “Sistema de Informação de Vigilância Epidemiológica da Gripe”).

In our study, our primary outcome was to describe the demographical and clinical characteristics of hospitalized individuals with CF affected by SARI due to another known respiratory virus than SARS-CoV-2, an unknown etiological agent, or SARS-CoV-2 (COVID-19). The secondary outcomes of the epidemiological study were: (i) to associate the demographical and clinical characteristics of hospitalized individuals with CF affected by SARI in the group of patients with COVID-19 with the outcomes (death or clinical recovery), and (ii) to characterize the virus from the patients with CF classified as SARI due to another known respiratory virus, as well as the co-detection in patients with positive SARS-CoV-2 infection.

We obtained the data from 29 December 2019 to 20 March 2022, and we evaluated the following information (patients’ features): (i) virus profiles that cause the SARI; (ii) demographical profile, including sex, age, education level, and place of residence; (iii) data for place of residence with flu outbreak, presence of hospital-acquired infection (nosocomial), and antiviral drug use to treat influenza virus infection; (iv) presence of comorbidities (cardiopathy, diabetes mellitus, systemic arterial hypertension, respiratory disorders, obesity, and others); (v) clinical symptoms related to SARI (fever, cough, loss of smell, loss of taste, myalgia, and others); (vi) the need for ICU and/or mechanical ventilation support (invasive or non-invasive); (vii) outcomes (death or clinical recovery); (viii) time between symptoms onset, hospitalization, and outcome; (ix) time in ICU; and (x) COVID-19 vaccination status (exploratory analysis). For accuracy, all epidemiological data from individuals with SARI and CF obtained in the dataset were revised by two authors (MNB and FALM). The definitions for symptoms and previous diseases were carried out by both authors (MNB and FALM).

All the patients included in the study had the diagnosis of CF and SARI; however, they were further divided into three groups, depending on the SARI etiological agent, as follows: (i) SARI due to another viral infection other than SARS-CoV-2 (e.g., influenza virus); (ii) SARI due to an unknown etiological agent; and (iii) SARI due to SARS-CoV-2 infection (patients with COVID-19). We followed the STROBE guideline for observational studies [23].

### 2.3. Statistical Analysis for the Epidemiological Study of the Brazilian Hospitalized Patients with CF and SARI

We performed the inclusion of missing data for some features because (i) we had more than 5% missingness, (ii) we identified the missing data only for the independent variable, and (iii) we assumed that the variables were missing completely at random. Also, we excluded the patients’ characteristics that had more than 40% missingness, which included the education level (56.1%), place of residence in a region with a previous flu outbreak (61.5%), abdominal pain (42.7%), fatigue and asthenia (40.8%), loss of smell (47.7%), loss of taste (47.2%), image exams results (X-ray (61.9%) and high-resolution computed tomography of the chest (79.3%)), and COVID-19 vaccination status (62.7%—this marker was excluded from the inferential statistical analysis protocol, but it was presented as part of the exploratory analysis). We imputed the missing data using the XLSTAT Statistical Software for Excel. For the qualitative (categorical data), we estimated the missing data using the NIPALS (Nonlinear Iterative Partial Least Squares) algorithm; also, for the quantitative (numerical data), we estimated the missing data using the MCMC (Markov Chain Monte Carlo) multiple imputation algorithm. The XLSTAT Statistical Software generated a new Excel dataset for descriptive and inferential statistical analyses.

We performed the statistical analysis in the Statistical Package for the Social Sciences software (IBM SPSS Statistics for Macintosh, Version 27.0.) and OpenEpi software (OpenEpi: Open-Source Epidemiologic Statistics for Public Health, Version. www.OpenEpi.com, (accessed on 19 June 2023)). The chi-square or Fisher’s exact statistical tests were used to compare the proportion of individuals with CF and SARI due to another viral infection than SARS-CoV-2, SARI due to an unknown etiological agent, or SARS-CoV-2 infection, considering all patients’ features evaluated in the present study. Also, the same statistical tests were used to calculate the chance of death among individuals with positive SARS-CoV-2 infection concerning patients’ features. The odds ratio (OR) and the 95% confidence interval (95%CI) were calculated using the chi-square or Fisher’s exact statistical tests for each analysis. The OR was calculated using the OpenEpi software for 2 × 2 tables, including the value for each patient characteristic. In addition, we compared the time between symptoms onset, hospitalization, and outcome, and time in ICU between the SARI groups (SARI due to another viral infection than SARS-CoV-2, SARI due to an unknown etiological agent, or SARS-CoV-2 infection), as well as according to the outcomes (death or clinical recovery) in the patients who presented positive for SARS-CoV-2 infection. We used the Kruskal–Wallis and Mann–Whitney tests to compare the times (numerical data—number of days) between the study groups. The results were summarized in tables and figures. The figures were built using the GraphPad Prism version 8.0.0 for Mac, GraphPad Software, San Diego, CA, USA (http://www.graphpad.com (accessed on 19 June 2023)).

## 3. Results

### 3.1. Systematic Review of COVID-19 among Those with CF—Worldwide Population

A total of 31 papers published as observational studies reported an association of COVID-19 in patients with CF [18,19,20,21,24,25,26,27,28,29,30,31,32,33,34,35,36,37,38,39,40,41,42,43,44,45,46,47,48,49,50] (Table 1). These studies comprised 661,386 patients in total, including children, adults, and elderly age groups. Also, pregnant women were described in two papers only [42,48]. However, only 19,150 (2.9%) patients were diagnosed with CF and, from these patients, 2523 (0.4%) were diagnosed with both CF and COVID-19. Regarding patients with CF and COVID-19, the sex proportion was (female/male) 1207/1167 [24,25,27,28,29,30,31,32,33,34,35,36,37,38,39,41,42,43,45,47,48,49,50]. Some publications [21,26,35,40,46] did not report the sex of the patients.

The most frequent symptom was fever, reported in 620 patients with CF [19,20,24,30,31,33,34,35,37,41,43,46,47,50], followed by cough (549 patients with CF) [24,31,33,34,41,42,46,47,50], and pulmonary symptoms (409 patients with CF) [33,34].

Other clinical symptoms were also described, like (number of patients with CF): fatigue (247) [31,33,46], myalgia (209) [19,20,33,41,43,47,50], headache (189) [20,31,33,41,43,46,48], dyspnea (245) [19,20,21,33,34,41,45,46], chest tight (45) [33], wheeze (15) [33,47], increased sputum (140) [21,31,33,41,42,46,50], respiratory failure (16) [31,33], diarrhea (70) [33,41,46,50], nausea (39) [33,41,43,46,48], abdominal pain (22) [33,46], joint pain (60) [20,31,41], anosmia (67) [33,41], dysgeusia (51) [33,41], pneumonia (5) [24,26,35], gastrointestinal symptoms (42) [19,24], acute respiratory failure (4) [26,41], asthenia (84) [41,43], chest pain (3) [41,50], rhinitis (3) [41,43,46], and limb pain (1) [46].

Regarding the *CFTR* genotype, the most common pathogenic variant was the p.Phe508del. In the studies, 623 patients were p.Phe580del heterozygous, then 486 patients were p.Phe580del homozygous, and, also, 496 patients presented other pathogenic variants [19,20,21,25,30,31,33,34,38,39,40,41,45,47,48,49,50]. A total of 300 patients with CF were lung transplanted; the remaining individuals were non-transplanted patients or did not have their organ transplantation specified [18,20,21,24,25,26,27,28,29,30,31,32,33,34,35,36,37,38,39,40,41,42,43,44,45,46,47,49,50]. 

Most of the included papers were conducted in European countries; Italy had the highest number of papers published with their population, which appears eleven times (35.5%) [18,19,20,25,33,35,40,41,45,47]. In addition, other European countries appeared fewer times (14, 45.2%) [18,19,21,25,31,32,33,34,35,38,39,46,48,49]. Other studies were also performed on populations from other nations, like the United States of America, Mexico, India, Oman, Brazil, Argentina, Chile, South Africa, Australia, and Canada [19,27,28,29,30,34,42,44,50].

Most of the diagnoses of COVID-19 in the included papers was made using the real-time polymerase chain reaction (RT-PCR) from nasopharyngeal swabs; however, some studies used the serology tool [37,39,46,50]. Computed tomography of the chest was used for complementary diagnosis to evaluate the pulmonary disease manifestation or evolution in five (16.1%) studies only [21,24,25,33,49].

Nine (29.0%) studies [31,33,36,39,40,41,46,47,50] reported asymptomatic cases; nevertheless, most of the patients were symptomatic [17,18,21,22,23,24,25,26,27,28,29,31,32,34,35,38,39,40,41,42,43,45,46,47]. The studies reported that children have a milder clinical presentation of COVID-19 and a better prognosis than adults, even with CF. In addition, only one paper reported an asymptomatic case of COVID-19 in a child [36]. Regarding adults, those with COVID-19 and CF who needed a lung transplant were observed to develop greater severity of COVID-19 than those who did not need a lung transplant [18,20,21,24,31,32,34,38].

Most of the papers described the outcomes of the patients with CF and COVID-19, and it was observed that the most common outcome was the need for hospitalization (n = 322 patients with CF) [19,20,21,25,27,28,30,31,32,33,34,35,38,44,45,50] and the need for oxygen support (n = 139 patients with CF) [19,20,25,28,31,33,34,35,45]. One hundred patients with CF needed ICU [19,21,25,26,27,28,29,30,31,32,33,34,38,48,49,50], fifty patients needed non-invasive mechanical ventilation support [19,20,21,25,33,44,50], and, curiously, only three patients were described as receiving invasive mechanical ventilation support [19,31,35]. In addition, one study described the phenotype of those four patients with CF who received extracorporeal membrane oxygenation respiratory support [33].

Deaths were observed in some papers (n = 38 patients with CF) [29,33,38]. Importantly, lung-transplanted patients with CF presented with an increased risk of death in one publication [38]; in accordance, another study described that lung transplantation and moderate to severe lung disease were independent risk factors for severe outcomes after SARS-CoV-2 infection [33]. It is worth mentioning that, in a study that included patients from several diseases, in which there were two deaths, it is not clear which patients died, so it was not counted in the sum of the cases of deaths [24].

In addition to CF, the studies have reported several other comorbidities, such as pulmonary hypertension, obesity, asthma, other chronic obstructive pulmonary disease, obstructive sleep apnea, and lung cancer [20,26,31,31]. But the most frequent diseases associated with CF were cystic-fibrosis-related diabetes (CF-RD) (n = 610 patients with CF) [20,24,29,31,32,33,34,38,41,50], pancreatic failure (n = 867 patients with CF) [20,33,38,41,45,49,50], systemic arterial hypertension (n = 268 patients with CF) [29,31,32,33,50], and chronic lower respiratory disease (n = 259 patients with CF) [29]. Other comorbidities were reported, such as liver disease (n = 77 patients with CF) [20,31,41,44,50], kidney disease (n = 106 patients with CF) [29,32,41,50], heart disease (n = 96 patients with CF) [29,31], and other types of cancer (n = two patients with CF) [32].

Other common etiological agents from CF were identified in addition to SARS-CoV-2, such as *Pseudomonas aeruginosa* (n = 613 patients with CF; one patient was using an anti-*P. aeruginosa* therapy) [19,20,21,24,28,31,33,38,41,44,45,47], *Staphylococcus aureus* (n = 498 patients with CF) [19,21,33,38,47], *Burkholderia cenocepacia* (n = 37 patients with CF) [33,35,38,41,47], *Stenotrophomonas* ssp. (n = 66 patients with CF) [33,47], *Nontuberculous mycobacteria* (n = ten patients with CF) [19], *Mycobacteria* ssp. (n = 28 patients with CF) [33], *Stenotrophomonas maltophilia* (n = one patient with CF) [38], and *Aspergillus* ssp. (n = 163 patients with CF) [21,26,28,31,33,41].

The pharmacological interventions reported included antibiotics, used in 1272 individuals, which included Azithromycin (n = 451 patients with CF) [27,31,33,41,47], oral antibiotics (n = 253 patients with CF) [19,33,50], intravenous antibiotics (n = 352 patients with CF) [19,33,50], and non-specified antibiotics administration (n = 216 patients with CF) [19,20,24,28,35,37,38,41,44,45,48,49]. An inhaled corticosteroid was prescribed to 119 patients with CF [20,31,37,41]; furthermore, an oral corticosteroid was used in 96 patients with CF [20,21,24,27,31,38,41,44,48,49]. In addition, salbutamol was used in only two patients with CF [28,40] and the dornase alfa in 82 patients with CF [21,30,41]. Curiously, antiviral drugs, such as Ritonavir [19,38], Lopinavir [20,38], and Remdesivir [27,44,49,50], were prescribed to 22 patients with CF. Some studies described therapy with monoclonal antibodies (mAbs) to treat patients with CF affected by COVID-19 [27,38,49]; in addition, one study described the use of immunotherapy with Omalizumab [27]. Hydroxychloroquine was administered to 16 patients with CF [20,27,35,38,50]. Telemedicine was used to reduce cross-infection and obtain clinical data for SARS-CoV-2 infection [39,47].

### 3.2. Demographic and Clinical Characteristics of the Brazilian Hospitalized Patients with CF and SARI

We assessed the data of 3,177,462 hospitalized patients due to SARI; we recovered the data of 501 patients with CF from them. We excluded 75 patients due to the absence of outcomes, 66 from the SARI due to an unknown etiological agent group, and nine with SARI due to the SARS-CoV-2 infection group. In this context, we enrolled a total of 426 hospitalized patients with CF and SARI during the period of the study. The distribution of the patients according to the place of notification (Figure 1) and state of residence is presented in Appendix A. In addition, we presented the cumulative number of cases for SARI due to an unknown etiological agent group and SARS-CoV-2 infection group by month of the COVID-19 pandemic considering the outcomes in Figure 2.

In our data, most of the patients with CF were female (230; 54.0%). Most of those included identified themselves as White (272; 63.8%), followed by *Pardos* (Multiracial background – 132; 31.0%), and Black (15; 3.5%). Individuals aged <5 years (96; 22.5%) and >50 years (119; 27.9%) were the most prevalent (Table 2). Those who lived in an urban region were the most common individuals (385; 90.4%), few patients had a nosocomial infection (7; 1.6%), and few individuals were treated with antiviral drugs (82; 19.2%), Oseltamivir being the most used drug (78; 18.3%) (Table 2).

The most common symptoms were cough (357; 83.8%), followed by dyspnea (355; 83.3%), and respiratory discomfort (334; 78.4%), whereas the most common comorbidities were cardiopathy (54; 12.7%), diabetes mellitus (46; 10.8%), and asthma (22; 5.2%) (Table 2). Almost one-third of the patients needed ICU treatment (135; 31.7%), and most of the patients in this cohort needed mechanical ventilatory support of some kind, invasive (67; 15.7%) or non-invasive (244; 57.3%) (Table 2). Most of the patients had SARI due to an unknown etiological agent (286; 67.1%), followed by SARI due to SARS-CoV-2 infection (117; 27.5%), and SARI due to another viral infection (23; 5.4%) (Table 2). The main discharge criterion was based on laboratory findings (400; 93.9%) rather than clinical findings (26; 6.1%). Finally, most patients had clinical recovery (319; 74.9%); however, several patients died due to SARI (94; 22.1%). The complete demographical and clinical characteristics are summarized in Table 2.

### 3.3. Demographic and Clinical Characteristics of Brazilian Hospitalized Patients with CF According to the SARI Group

As mentioned previously, our study accounted for three groups of patients with CF and SARI, namely, SARI due to another viral infection than SARS-CoV-2, SARI due to an unknown etiological agent, and SARI due to SARS-CoV-2 infection. In all three groups, most of the individuals were White, with a higher prevalence of this race in individuals with SARI due to an unknown etiological agent (195; 68.2%) (*p* = 0.026) (Table 3). Regarding age, individuals with SARI due to SARS-CoV-2 infection tended to be older, especially over 50 years old (63; 53.8%), whereas, for SARI due to another viral agent than SARS-CoV-2 (16; 69.6%) and SARI due to an unknown etiological agent (70; 24.5%), a higher prevalence of individuals under five years old was observed (*p* < 0.001). Although in all three groups, most of the patients lived in an urban area, a higher percentage of individuals living in a peri-urban region was observed in individuals with SARI due to another viral infection than SARS-CoV-2 (3; 13.0%) when compared to the other groups (*p* = 0.024) (Table 3). There was no difference between the three groups regarding sex, nosocomial infection, antiviral drug use to treat the flu symptoms, and discharge criterium (Table 3).

A higher prevalence of individuals with SARI due to SARS-CoV-2 infection presented (107; 91.5%) dyspnea when compared to SARI due to another viral agent than SARS-CoV-2 (20; 87.0%) and SARI due to an unknown etiological agent (228; 79.7%) (*p* = 0.010) (Appendix A; Figure 3A). In the same way, individuals with SARI due to SARS-CoV-2 infection (95; 81.2%) also presented a higher prevalence of lower peripheral arterial oxygen saturation (SpO_2_) when compared to the other groups [SARI due to another viral agent than SARS-CoV-2 (14; 60.9%) and SARI due to an unknown etiological agent (202; 70.6%)] (*p* = 0.035) (Appendix A; Figure 3A). Finally, cardiopathy was more common in individuals with SARI due to SARS-CoV-2 infection (32; 27.4%) when compared to the other groups [SARI due to another viral agent than SARS-CoV-2 (no cases) and SARI due to an unknown etiological agent (22; 7.7%)] (*p* < 0.001). There was no statistical difference for other symptoms or comorbidities (Appendix A; Figure 3A,B).

The highest prevalence for ICU treatment was for patients with SARI due to SARS-CoV-2 infection (52; 44.4%) (*p* < 0.001), as well as the need for invasive mechanical ventilation (29; 24.8%) (*p* = 0.007; Appendix A; Figure 4). Almost half of the patients with SARS-CoV-2 infection and CF died (51; 43.6%); in contrast, none of the patients classified in the SARI due to another viral infection group died. However, we observed 43 (15.0%) deaths due to SARI in the group of patients classified as SARI due to an unknown etiological agent (*p* < 0.001; Appendix A; Figure 4). In addition, 12 (4.2%) deaths and one (0.9%) death not associated with the SARI occurred, respectively, in the groups classified as SARI due to an unknown etiological agent and SARI due to COVID-19. Furthermore, individuals with SARI due to an unknown etiological agent were hospitalized for more days (14.53 ± 18.4 days) (Table 4). In contrast, individuals with SARI due to SARS-CoV-2 infection were hospitalized for more days in ICU (2.31 ± 4.31 days) (Table 4). All data are summarized in Appendix A, Figure 4, and Table 4.

In addition, we performed an exploratory analysis regarding the response (outcomes) for COVID-19 vaccination in the groups of hospitalized patients with CF classified as SARI groups by outcomes. The COVID-19 vaccination status was described for few individuals. In our data, no hospitalized patients classified as SARI due to another viral infection group died and only 1/15 (6.7%) was vaccinated (Table 5). Among those classified as SARI due to an unknown etiological agent, 32/94 (34.0%) patients were COVID-19 vaccinated and 5/11 (45.5%) deaths occurred among the vaccinated ones (*p* = 0.390) (Table 5). Among those classified as SARI due to SARS-CoV-2 infection, 26/50 (52.0%) patients were COVID-19 vaccinated and 14/20 (70.0%) deaths occurred among the vaccinated ones (*p* = 0.139) (Table 5). Finally, in the pooled analysis, 59/159 (37.1%) patients were COVID-19 vaccinated and 19/31 (61.3%) deaths occurred among the vaccinated ones (*p* = 0.041) (Table 5). 

### 3.4. Clinical and Demographic Characteristics Associated with Death in Brazilian Hospitalized Patients with CF and COVID-19 

A total of 51/116 (44.0%) patients with both CF and COVID-19 died, and 65/116 (66%) recovered during the study. A higher prevalence of death was observed with older age, especially those over 50 years old (37; 72.5%); in this context, we compared the chance of death among the patients aged above and below 50 years, and we observed an OR of 3.964 (95%CI = 1.799 to 8.736) for those aged over 50 years (Table 6). Most of those who died had SpO_2_ <95% (46; 90.2%—OR = 3.258 (95%CI = 1.111 to 9.556)) whereas most of those who recovered presented with a fever (47; 72.3%—OR = 0.368 (95%CI = 0.170 to 0.797)) (Appendix A; Figure 5A). Finally, most of the patients who died needed ICU treatment (34; 66.7%—OR = 5.647 (95%CI = 2.530 to 12.610)) and needed invasive mechanical ventilation (27; 52.9%—OR = 82.700 (95%CI = 12.480 to 1073)) when compared to those who recovered (Appendix A; Figure 6). There was no statistical significance regarding sex, race, place of residence, nosocomial infection, other clinical symptoms, comorbidities, antiviral use, or discharge criterion with the outcome (Table 6; Appendix A; Figure 5A,B).

Finally, individuals with SARI due to SARS-CoV-2 infection who died were hospitalized for more days in ICU than those who presented clinical recovery (3.86 ± 5.66 days vs. 1.10 ± 2.27 days; *p* < 0.001) (Table 4).

### 3.5. Microbiological Profile of Brazilian Hospitalized Patients with CF According to the Three Groups of SARI

Several viruses other than SARS-CoV-2 were identified in our cohort; for instance, in individuals with SARI due to a viral infection other than SARS-CoV-2, the most common viruses were rhinovirus (12 cases), followed by respiratory syncytial virus (six cases), parainfluenza 3 (four cases), influenza A (two cases), bocavirus (two cases), parainfluenza 1 (one cases) and parainfluenza 4 (one case). Regarding SARI due to an unknown etiological agent, in most patients, the RT-PCR was negative for a viral infection (263 cases); however, some patients did not have the RT-PCR result (23 cases). In the group of patients with COVID-19, the SARS-CoV-2 infection was confirmed by RT-PCR, and infections with rhinovirus (two cases, no deaths) and influenza A (one case who died) were also diagnosed (Table 7).

## 4. Discussion

To the best of our knowledge, this study is the only one that comprised a uniquely Brazilian cohort of hospitalized patients with both CF and SARI. We further divided our cohort into three different groups, namely, SARI due to another viral infection than SARS-CoV-2 ones (23; 5.4%); SARI due to an unknown etiological agent (286; 67.1%), and SARI due to SARS-CoV-2 infection (117; 27.5%) to tease apart, among the etiological agents that cause SARI, the effect that SARS-CoV-2 has on CF individuals in Brazil. Patients in the latter group seemed to have a more severe clinical condition, since they presented a higher case fatality rate, need for ICU, need for invasive mechanical ventilation, and even a higher prevalence of clinical symptoms related to higher severity, such as dyspnea and low peripheral oxygen saturation, when compared to the other study groups.

Previous studies reported the clinical characteristics and outcomes of individuals with CF and SARS-CoV-2 [18,19,20,21,24,25,26,27,28,29,30,31,32,33,34,35,36,37,38,39,40,41,42,43,44,45,46,47,48,49,50]. Unfortunately, most of the studies were performed in the European continent, where the genetic variability in the *CFTR* gene is low. For example, one European collaborative study identified homozygous and heterozygous individuals for the p.Phe508del pathogenic variant in the *CFTR* gene in ~46.0% and ~39.0% of the individuals, respectively [51]. The Brazilian population, however, is extremely mixed due to intense miscegenation between African, native Indigenous, and European people, which consequently leads to high genetic heterogeneity, as observed in the different genetic profiles of patients with CF, in which p.Phe508del, p.Gly542X, p.Asn1303Lys, p.Gly551Asp, and p.Arg553X pathogenic variants of the *CFTR* gene were observed in only ~56.0% of CF reported cases, with a wide variation across the Brazilian states [52,53,54,55,56]. Thus, researchers must describe the outcomes of COVID-19 in patients with CF with different genetic profiles, like in the case of Brazil, where rare and new genetic variants in the *CFTR* gene were previously described [57,58,59,60].

In the study performed by Hadi et al. (2021) [29], 422 patients with CF and COVID-19 were included from more than 40 health care centers in the United States of America, and most of the patients included were female and Caucasian (White individuals), similar to our study. The authors observed enhanced 30-day mortality (5.2% vs. 1.7%), ICU admission (11.6% vs. 2.6%), and need for mechanical ventilation (6.2% vs. 1.6%) in patients with COVID-19 and CF when compared to those with only COVID-19. In our study, however, we observed a higher case fatality rate, need for mechanical ventilation, and ICU care admission than reported by Hadi et al. [29], perhaps due to the fact that all of our patients needed hospitalization, and only 117 (23.3%) of Hadi et al.’s cohort needed inpatient service, which may correspond to the more severe clinical condition of our cohort.

A French study included 31 patients with COVID-19 and CF [31]. Regarding the distribution of symptoms, the authors observed fever, cough, and fatigue to be the most common ones, and only 19 (61.3%) of them needed hospitalization, whereas, in our study, cough, dyspnea, and respiratory discomfort were the most frequent and all the patients needed hospitalization. Perhaps this difference can be ascribed to the patient selection workflow, which included a more severe clinical condition in our patients [31]. Similar to our study, high ICU admission was observed; however, only one (3.2%) patient with CF needed invasive mechanical ventilation, and none of them died [31]. Another study carried out in Europe, in Italy, by Colombo et al. (2022) [41] comprised 236 patients with CF and positive RT-PCR for SARS-CoV-2. Most patients reported fever, cough, and asthenia as the most common symptoms, whereas 22.0% were asymptomatic [41]. Only 43 (18.2%) patients required hospital admission, and four (1.7%) required ICU treatment. Also, six (2.5%) patients with CF died [41]. Finally, one of the most extensive studies published in Europe comprised 828 patients with CF and COVID-19. The most common symptoms were cough, fever, and fatigue, but only 586 (75.7%) of them reported any symptoms [33]. A total of 195 (23.7%) patients with CF required hospitalization, 32 (3.9%) required respiratory support, of which 12 (1.5%) needed invasive ventilation; only 21 (2.5%) patients needed ICU care, and 11 (1.4%) patients with CF died [33]. One of the main reasons our results differ from the current literature, that is, our results show an enhanced need for ICU, need for invasive mechanical ventilation, and case fatality rate, might be the fact all the patients from our cohort were hospitalized, which translates into a more severe clinical condition.

Only two studies reported patients with CF and COVID-19 in Brazil [19,21]. Bain et al.’s study (2021) [19] comprised 105 children from 13 countries, such as Argentina, Chile, Brazil, France, Germany, Italy, Russia, South Africa, Spain, Sweden, Switzerland, the United Kingdom, and the United States of America. Unfortunately, the authors could not detail the patients’ nationalities due to the need to maintain confidentiality. A total of 24 out of 82 (29.3%) patients needed to be hospitalized, and only one out of 83 (1.2%) needed ICU care. Regarding respiratory support, six out of 21 (28.6%) needed supplemental oxygen, two out of 20 (10.0%) needed non-invasive mechanical ventilation, and one out of 20 (5.0%) needed invasive mechanical ventilation [19]. These results contrast with ours; our patients needed hospital treatment, demonstrating an enhanced severity that can explain our results. The authors also observed several differences between hospitalized and non-hospitalized patients, for instance, those with higher body mass index, pancreatic insufficiency, worse percent predicted forced expiratory volume in one second, and those not using modulator treatment appear to need more hospitalizations [19].

The other study with a Brazilian patient was a case report in which the authors described an infection by SARS-CoV-2 in a 37-year-old male patient with CF in the second week after lung transplantation [21]. On the 9th post-operative day, the patient developed acute rejection, being treated with a 3-day course of methylprednisolone on the 11th post-operative day [21]. However, the patient still presented hypoxemia; after a computerized tomography showed ground glass opacities, a COVID-19 nasopharyngeal swab was performed. The patient rapidly progressed to severe hypoxemia, needing mechanical ventilation support, in which the tracheal aspirate was positive for SARS-CoV-2 and adenovirus. On the 19th post-operative day, he was weaned off mechanical ventilation support. On the 39th post-operative day, he was discharged from the hospital, even though the patient was still positive for COVID-19 15 and 25 days after symptom onset [21]. Unfortunately, there are no other studies comprising a Brazilian cohort of patients with CF and COVID-19, and no studies consisting of a cohort of adult patients, which makes our study one of the first, and necessary since the Brazilian population is highly mixed and has some unique traits, as previously mentioned.

Several other pathogens commonly colonize individuals with CF; in fact, several bacteria such as *S. aureus*, *Haemophilus influenzae*, *P. aeruginosa*, and *Burkholderia complex* have been widely identified in individuals with CF, being that the most common respiratory pathogen in adults is *P*. *aeruginosa,* whereas, in children, it is the *S. aureus* [61,62]. Although individuals with COVID-19 tend to have a relatively low rate of co-infection with bacterial agents, *S. pyogenes* and *S. pneumoniae* being the most common microorganisms [63,64], it might not be true for patients with CF and COVID-19, since nearly 25.0% of the patients included in the systematic review presented co-infection of *P. aeruginosa* and SARS-CoV-2 [19,20,21,24,28,31,33,38,41,44,45,47], which might impair their prognosis. Unfortunately, in our data, we did not have access to bacterial testing data because the Brazilian Ministry of Health did not compute the data into the dataset. Regarding viral infections, the most common viruses identified were the RNA-virus, being that respiratory syncytial virus, rhinovirus, influenza, and parainfluenza were the most common [65,66,67,68]. In addition, the data regarding the co-infection of SARS-CoV-2 and other viruses are scarce in the literature, e.g., only one study reported a co-infection of SARS-CoV-2 and adenovirus in a CF lung-transplanted patient [21]. However, patients with CF and COVID-19 seemed to have the same virus profile as other patients, since the most common virus identified in our cohort, rhinovirus, was also the most commonly identified virus in a cohort of no CF infected with SARS-CoV-2 [63]. One might speculate that the low incidence of other viral agents might be due to the adherence to social isolation observed in patients with CF since they are worried about their respiratory health. However, it is essential to evaluate how much COVID-19 and the measures to control its progression, such as social isolation, impacted the patients with CF, including psychologically [63,69,70].

The association between CF and viral infections is not new in the literature; reports have shown that up to ~60.0% of acute pulmonary exacerbations in patients with CF are caused by viral infections [65,71,72]. Although previous studies observed worse outcomes in patients with CF and other viral infections, especially influenza and respiratory syncytial virus [13,72,73,74], the same does not seem to happen with individuals with COVID-19 [34,36,72,74]. Several unique traits of CF might help explain this phenomenon; for instance, these patients might have paid more attention to respiratory infection, respecting social distance, and infection control, which might explain the lower incidence of COVID-19 in patients with CF [34,72,74]. Furthermore, in a recent analysis, increased angiotensin-converting enzyme-2 (ACE2) mRNA, and decreased transmembrane protease serine 2 (TMPRSS2) mRNA, were described in CF airway epithelial cells when compared to non-CF cells [72,75,76], both of which are important proteins for the SARS-CoV-2 entry in cells. Although the increased ACE2 expression might increase the SARS-CoV-2 binding to epithelial cells, it is also responsible for converting angiotensin II to angiotensin 1–7, an anti-inflammatory molecule, which might decrease pulmonary damage from the SARS-CoV-2 and inflammation [72,77,78]. Also, the low expression of TMPRSS2 might reduce the SARS-CoV-2 entry in cells [72]. The increased ACE-2 and decreased TMPRSS2 expressions might explain the lower impact of SARS-CoV-2 in patients with CF; however, our study contrasts with the present literature because our patients with CF and COVID-19 presented worse outcomes than those infected by other viruses.

Our study also shows some unique traits from our patients that might help explain this finding: for instance, as previously mentioned, all of our patients with CF were hospitalized, with greater severity of SARI, whereas most of the patients from other studies were outpatients; most of the referral centers to manage CF in Brazil are located in the Southeast region, especially São Paulo state [79], which can complicate the logistics of these patients, especially when in relation to seeking proper medical care; and, although available in the Brazilian free health care system, most of the drugs used to treat CF, such as alpha-dornase and Ivacaftor [80], are expensive and represent an impact in the healthcare budget [81], which limits the number of people who can use them, especially at present, since the Brazilian Ministry of Health had a budget cut of more than 40.0% [82]. All of this might contribute to worse outcomes in this population and perhaps explain, in part, our findings.

Finally, several measures taken to mitigate the impact of the COVID-19 pandemic, such as social isolation, might have a negative impact on patients with CF, especially in relation to physical exercise [83]. Recent studies observed that patients with CF were doing less physical activity (ranging from 45.0 to 71.0%), mostly because the fitness center was closed or even due to lack of motivation [32,84,85]. The reduction in physical exercise is troublesome for patients with CF, since physical activity is part of the treatment and is responsible for decreased hospitalizations and a slower rate of lung function decline [32,86,87,88]. Luckily, most patients are willing to change their exercise routine and implement online classes to keep doing physical exercises [85].

Our study comprised only Brazilian hospitalized patients with CF, which might present greater severity. We observed that their burden is enhanced in the COVID-19 pandemic, with worse outcomes, such as higher case fatality rates, need for ICUs, and need for mechanical ventilation support. However, the impact of COVID-19 is not limited to the disease itself since isolation proposed to decrease the SARS-CoV-2 spread might have also reduced their level of physical activity, thus impairing their treatment and lung disease progression, as well as psychological status.

Regarding COVID-19 immunization, many articles were carried out before the start of or during studies involving vaccines. Therefore, they are not precise as to the vaccine status of the sample [20,21,24,25]. Four papers do not presented the number of vaccinated individuals, however, they mention the need to prioritize patients with CF during campaigns [18,19,26,33]. Some papers do present the number of vaccinated patients; however, they do not describe whether they are part of the patients with CF and whether they presented with a positive result for SARS-CoV-2. An article [41], which was performed in Italy in mid-March 2021, cites the beginning of the vaccination campaign in the country, and although no participant in the paper had been immunized, it presents as a given a decrease in the contamination of the disease that can be the result of mass vaccination.

Limitations: In our study, we used a public dataset and did not have access to the original data. There is considerable underreporting of COVID-19 in Brazil, and several cases diagnosed as SARI due to unknown etiological agents might be COVID-19, as we presented in other studies [89,90,91]. Most of the referral centers to treat and follow patients with CF are in the South and Southeast regions of Brazil, making it difficult to manage patients from the North and Northeast regions. We did not have access to the *CFTR* genetic profile of patients with CF, which might have strengthened our work. Although we performed a systematic review, we did not perform a meta-analysis since the studies were very different, making a comparison impossible.

Further studies should evaluate the influence of modifier genes on the CF outcomes regarding COVID-19 because the modifier genes, including the variants in the *ACE* gene, were previously associated with the worst lung disease and response to treatment, including the response to personalized and precision medicine [92,93,94,95,96]. Our study is a picture coming from the Brazilian context and this could or could not match exactly the worldwide scenario, including access to the health system and/or the standard of admissible patients to treatment in ICU, together with code-status regulations. Also, in the future, it is important to perform other observational studies, such as those performed by the National Institute for Health and Care Research Global Health Unit on Global Surgery and COVIDSurg Collaborative (https://globalsurg.org/covidsurg/ (accessed on 19 June 2023)), in order to improve the world’s capacity to deal with conditions such as the COVID-19 pandemic (before, during, and after the COVID-19 pandemic) [97,98,99,100,101].

## 5. Conclusions

Individuals with CF might have an increased burden of COVID-19 compared to the general population, not only from the disease itself but also from public health policies to decrease the spread of SARS-CoV-2, such as social isolation, since their ability to engage in physical exercise decreases. Even though most of the other studies did not observe worse outcomes in patients with CF and COVID-19, our cohort comprised only hospitalized patients with greater severity of the disease, which might explain, at least in part, our findings of enhanced case fatality rate, need for ICU, and need for ventilatory support. Finally, our study is unique since it comprises only hospitalized Brazilian patients of all ages and with other causes of SARI.

## Figures and Tables

**Figure 1 healthcare-11-01936-f001:**
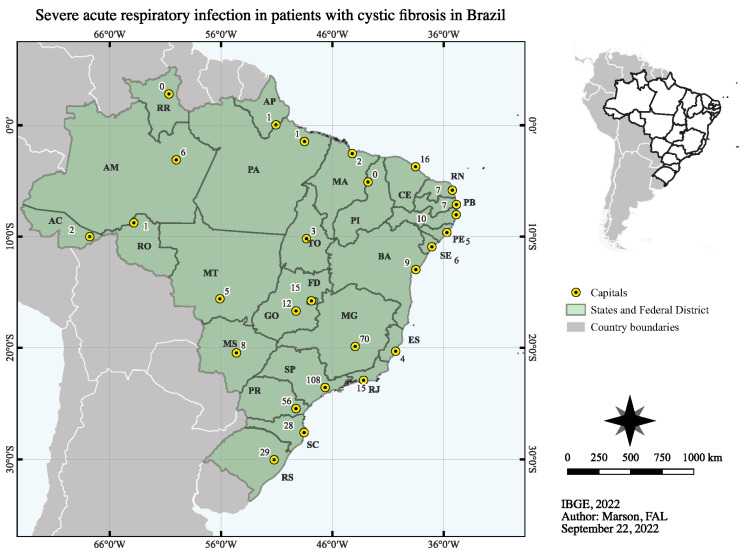
Distribution of the hospitalized patients with cystic fibrosis due to severe acute respiratory infection in Brazil during the coronavirus disease, COVID-19 pandemic. We showed the Brazilian states and Federal District (FD). We presented the data using the number of individuals (N). AC, Acre; AL, Alagoas; AM, Amazonas; AP, Amapá; BA, Bahia; CE, Ceará; ES, Espírito Santo; GO, Goiás; MA, Maranhão; MS, Mato Grosso do Sul; MT, Mato Grosso; MG, Minas Gerais; PA, Pará; PB, Paraíba; PA, Paraná; PE, Pernambuco; PI, Piauí; RJ, Rio de Janeiro; RN; Rio Grande do Norte; RS, Rio Grande do Sul; RR, Roraima; RO, Rondônia; Santa Catarina; SP, São Paulo; SE, Sergipe; TO, Tocantins; IBGE, Instituto Brasileiro Geografia e Estatística.

**Figure 2 healthcare-11-01936-f002:**
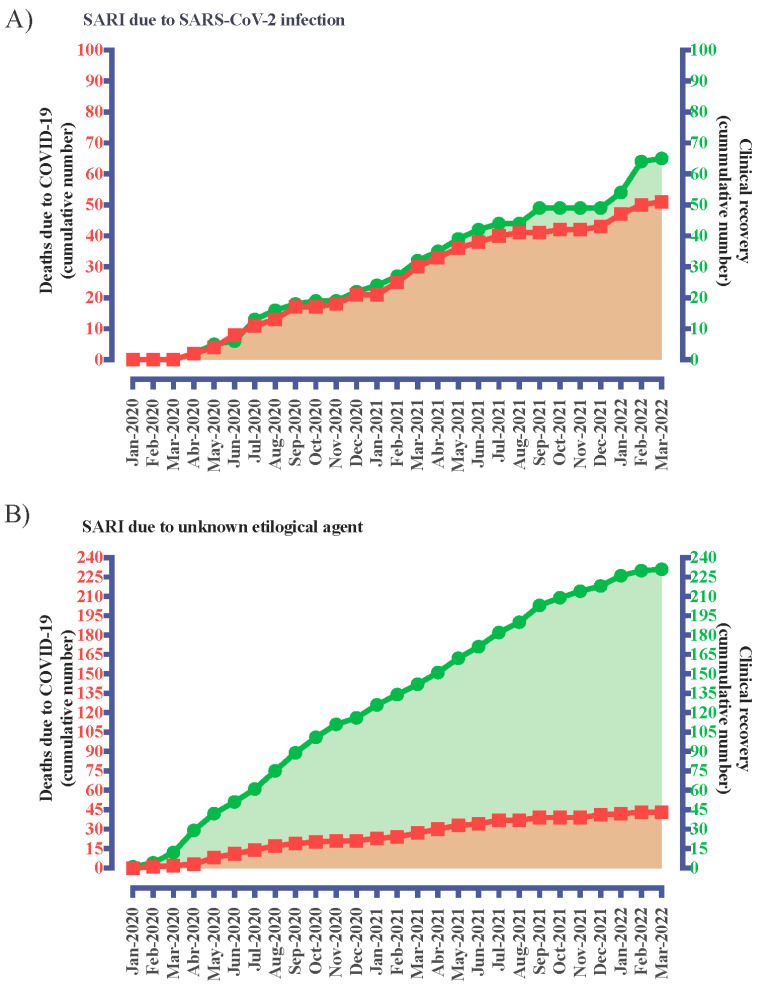
Cumulative number of cases for severe acute respiratory infection (SARI) due to an unknown etiological agent group (**A**) and severe acute respiratory syndrome coronavirus 2 (SARS-CoV-2) infection (**B**) group by month of coronavirus disease, COVID-19 pandemic considering the outcomes (number (n) of deaths and number of clinical recoveries). We obtained the data from 29 December 2019 to 20 March 2022.

**Figure 3 healthcare-11-01936-f003:**
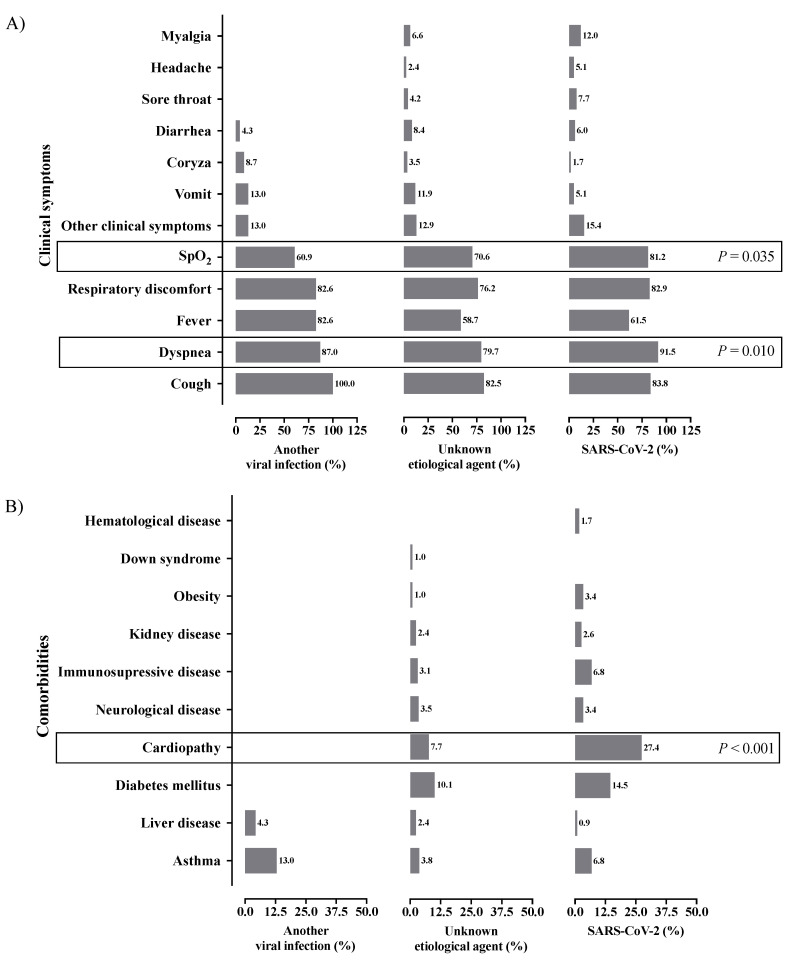
Description of the clinical symptoms and comorbidities of the hospitalized patients with cystic fibrosis according to the severe acute respiratory infection groups in Brazil during the coronavirus disease, COVID-19 pandemic. (**A**) Clinical symptoms; (**B**) comorbidities. We presented the data using the percentage (%). We presented the *p* for the significant association only and performed the statistical analysis using the chi-square test or Fisher’s exact test. We adopted an alpha error of 0.05. SARS-CoV-2, severe acute respiratory syndrome coronavirus 2; %, percentage; SpO_2_, peripheral arterial oxygen saturation.

**Figure 4 healthcare-11-01936-f004:**
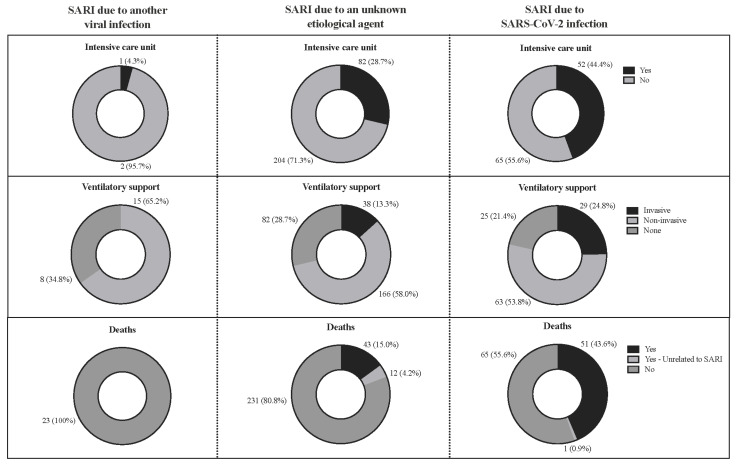
Association between the need for an intensive care unit, the need for mechanical ventilation, and the death of the hospitalized patients with cystic fibrosis according to the severe acute respiratory infection (SARI) groups in Brazil during the coronavirus disease, COVID-19 pandemic. *p* for the need for an intensive care unit = < 0.001; *p* for the need for mechanical ventilation = 0.007; *p* for death = < 0.001. We presented the data using the number of patients (N) and percentage (%). We performed the statistical analysis using the Fisher’s exact test. We adopted an alpha error of 0.05. SARS-CoV-2, severe acute respiratory syndrome coronavirus 2; %, percentage.

**Figure 5 healthcare-11-01936-f005:**
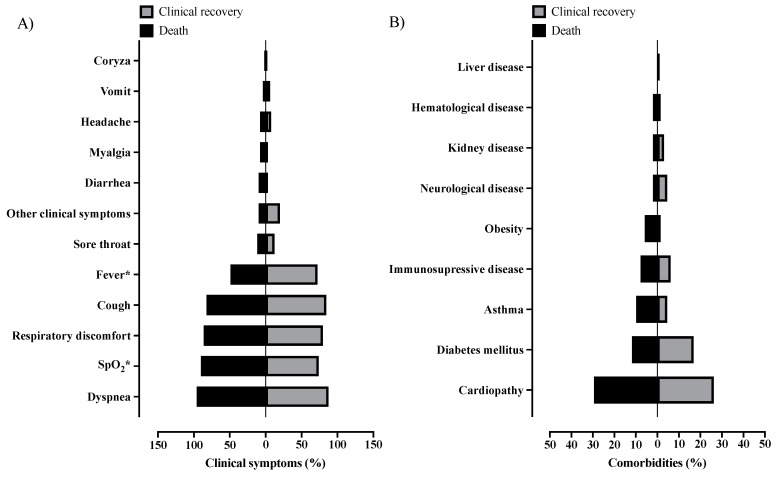
Association between clinical symptoms and comorbidities of the hospitalized patients with cystic fibrosis and SARS-CoV-2 infection in Brazil according to the outcome during the coronavirus disease, COVID-19 pandemic. (**A**) Clinical symptoms; (**B**) comorbidities. We presented the data using the percentage (%). We performed the statistical analysis using the chi-square test and Fisher’s exact test. We adopted an alpha error of 0.05. *, *p* with a significant value. SARS-CoV-2, severe acute respiratory syndrome coronavirus 2; %, percentage; SpO_2_, peripheral arterial oxygen saturation; OR, odds ratio; 95%CI, 95% confidence interval. Fever—OR = 0.368 (95%CI = 0.170 to 0.797); SpO_2_—OR = 3.258 (95%CI = 1.111 to 9.556).

**Figure 6 healthcare-11-01936-f006:**
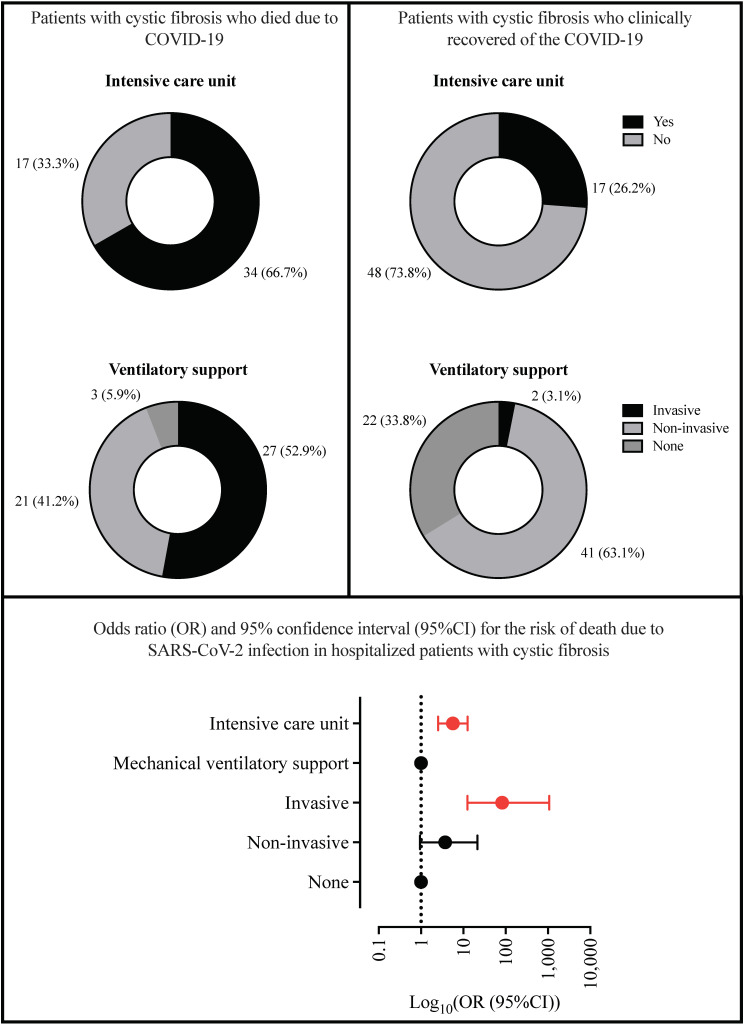
Association between the need for intensive care unit and mechanical ventilatory support with risk of death in hospitalized patients with cystic fibrosis and SARS-CoV-2 infection in Brazil during the coronavirus disease, COVID-19 pandemic. We presented the data using the number of patients (N) and percentage (%). We performed the statistical analysis using the chi-square test for the mechanical ventilatory support and Fisher’s exact test for the intensive care unit. We adopted an alpha error of 0.05. We presented the significant *p* value as red color. SARS-CoV-2, severe acute respiratory syndrome coronavirus 2; %, percentage.

**Table 1 healthcare-11-01936-t001:** Information regarding authorship, journal of publication, aim, sample, study type, methods, results, and conclusions of the studies included in the systematic review about the coronavirus disease, COVID-19 prevalence, manifestations, and outcomes in patients with cystic fibrosis (CF) [18,19,20,21,24,25,26,27,28,29,30,31,32,33,34,35,36,37,38,39,40,41,42,43,44,45,46,47,48,49,50].

Ref.	Journal of Publication	IF	Aim	Study Type	Sample	Methods	Results	Conclusions
[24]	*Transplant Infection Disease*	2.23	To describe the clinical course of eight patients with a history of lung transplant and infected with SARS-CoV-2.	Serial case report	Eight patients [five patients with idiopathic pulmonary, two patients with chronic obstructive pulmonary disease, and one patient with CF].	To describe the profile of eight patients who received lung transplantation but only five who had a positive RT-PCR for COVID-19. Among the patients, one had a CF diagnosis, which was the cause of the lung transplant.	Two patients needed mechanical ventilation, one of whom was prone. Two patients required a high-flow nasal cannula, one patient a regular nasal cannula, and three patients were not on supplementary oxygen (ambient air). The transplanted patients who died two weeks before the diagnosis were extubated post-transplant; however, they required mechanical ventilation after the COVID-19 diagnosis. The two patients who died developed sepsis or necrotizing pancreatitis. Of the surviving patients, six were discharged with stable graft function.	Short-term mortality of 25% was observed. Both patients who received Brasiliximab as an immunosuppressant died. In addition, 75% of patients who recovered from COVID-19 had their lung function preserved. The study did not describe its findings according to the disease that was the cause of lung transplantation.
[25]	*Journal of Cystic Fibrosis*	5.53	To assess the impact of SARS-CoV-2 infection in patients with CF.	Observational study—cohort	One hundred eighty-one patients with CF (32 post-transplant).	To present the outcome of COVID-19 in patients with CF from 19 European countries. For the diagnosis of COVID-19, positive results from material collected from the nasopharynx or throat and/or computed tomography of the chest and/or symptomatic hospital diagnosis were considered. The cases were subdivided according to the transplant of the lungs.	Transplanted patients with CF presented a higher proportion of people who needed hospitalization when compared with non-transplanted. Post-transplanted patients needed to be frequently admitted to intensive care units (7 vs. 4 patients) and required more supplementary oxygen and non-invasive ventilation than other patients. Also, the male sex was associated with severe disease, but the non-transplant CF cohort did not increase chances of hospitalization when compared with non-transplanted females. In the post-transplanted cohort, the male sex was twice as often hospitalized when compared to the female.	SARS-CoV-2 infection is less severe than feared in patients with CF; also, individuals with CF have median age with a lower prevalence of obesity and other factors. There was a propensity for hospitalizations for patients transplanted with CF when compared to non-transplanted ones. In terms of mortality, seven patients died—four non-transplanted patients died, and three were transplanted.
[19]	*Journal of Cystic Fibrosis*	5.53	To evaluate the cases of SARS-CoV-2 infection in children with CF under 18 years.	Observational study	One hundred and five children with CF.	The Registry Global Harmonization Group collected cases of children with CF in 13 European countries. For inclusion, positive SARS-CoV-2 RT-PCR and/or hospital clinical diagnoses were considered, and patients with antibody tests or self-reports were excluded.	The average age was ten years old, 54.0% of the patients were male, and the median percentage of the predicted forced expiratory volume in one second was 94.0%. About 71.0% of children were treated in the community. Of the admissions (n = 24), six patients required oxygen support and two non-invasive mechanical ventilation. Nearly half of the patients received antibiotic-type medications, five received antiviral drugs, four received Azithromycin, and one received corticosteroids. Lower lung function and body mass index were observed in hospitalized children. One child died from complications unrelated to SARS-CoV-2 infection six weeks after COVID-19 diagnosis.	The study provides information that will guide children with CF, their families, and the team responsible for care. It is noted that SARS-CoV-2 infection in children presents with mild symptoms in those who do not have pre-existing severe pulmonary involvement. The study considered children with CF as a priority group for access to the vaccine. It also emphasizes that patient follow-up is essential to assess long-term complications, including the impact on lung function.
[26]	*European Respiratory Journal*	33.80	To describe the clinical features of patients with prior lung disease who needed hospitalization for COVID-19 or 2018–2019 seasonal influenza and compare the pulmonary complications, need for intensive care unit, and in-hospital mortality according to both diseases.	Observational study—cohort	Patients with COVID-19 = 89,530 (chronic respiratory failure = 1433; chronic obstructive pulmonary disease = 4866; pulmonary hypertension = 341; asthma = 3273; CF with pulmonary manifestations = 20; lung cancer = 977; emphysema = 1426; pulmonary sarcoidosis = 159; interstitial lung disease = 1611; and sleep apnea = 3581). Patients with seasonal influenza = 45,819 (chronic respiratory failure = 1830; chronic obstructive pulmonary disease = 4637; pulmonary hypertension = 247; asthma = 2230; CF with pulmonary manifestations = 79; lung cancer = 431; emphysema = 533; pulmonary sarcoidosis = 81; interstitial lung disease = 471; and sleep apnea = 1443).	This nationwide retrospective cohort study describes patients with prior lung disease hospitalized for COVID-19 or influenza. The study compared the data for pulmonary complications, need for intensive care, and in-hospital mortality among the type of infection (virus) and groups of prior lung diseases.	Of the 89,350 cases of COVID-19 that made up the study, only 20 were individuals with CF and pulmonary manifestations. However, the number is less frequent when compared to seasonal flu cases (n = 45,819) in patients with CF (n = 79). There was an underrepresentation of patients with chronic respiratory failure, chronic obstructive pulmonary disease, asthma, CF, and pulmonary hypertension. Patients with lung cancer, sleep apnea, emphysema, and interstitial lung diseases were highly represented. Patients with COVID-19 and chronic respiratory diseases had a greater significance in the development of pneumonia and pulmonary embolism when compared to influenza, requiring intensive care more frequently. Except for patients with asthma, patients with chronic respiratory diseases with COVID-19 had higher mortality rates when compared to those with flu.	Patients with chronic respiratory diseases, especially asthma, chronic obstructive pulmonary disease, and chronic respiratory failure, were underreported among patients with COVID-19 when compared to those hospitalized with influenza. Chronic respiratory diseases are a risk for developing a more severe SARS-CoV-2 infection with an accompanying high mortality rate. The low incidence of the disease in patients with chronic respiratory disease can be explained by the great rigor with which they comply with social distancing. Regarding CF, the low incidence can be explained by the frequent use of inhaled corticosteroids that could protect patients against COVID-19. There was no need for intensive care, presence of exacerbations, or increased risk for individuals with CF and COVID-19 when compared to the other chronic respiratory diseases evaluated in the study.
[20]	*Plos One*	3.572	To describe the clinical signs and outcomes of SARS-CoV-2 infection in patients with CF from Italy.	Observational study–cohort	Patients with CF from 32 reference centers = 6597; 30 patients had suggestive symptoms of COVID-19, and 16 presented a positive diagnostic test. The 14 patients who tested negative for COVID-19 were classified as the control group.	The centers were contacted to collect data from patients who reported symptoms suggestive of COVID-19 after contacting positive or suspected third parties between February and July 2020. Symptoms and clinical outcomes were compared among patients who tested positive in nasopharyngeal swabs submitted to SARS-CoV-2 RT-PCR and the control group (negative RT-PCR). The clinical course of the infection was based on the need for intensive care, vital signs, supplemental oxygen, and mechanical ventilation support (invasive or non-invasive).	The most frequently reported symptoms were cough, fever, asthenia, and dyspnea; however, no statistical difference occurred in symptoms and clinical evolution between the two groups. Two adults with a history of lung transplantation required non-invasive ventilation, none required intensive care unit admission, and all progressed to cure without significant deleterious effects.	The clinical outcome of SARS-CoV-2 infection does not differ from other respiratory tract infections in CF. The clinical evolution of the infection in people with CF was less severe than expected, and this fact can be explained by the young age of the patients, the lower incidence of cardiovascular risk factors, and the presence of the respiratory microbiota that may lead to a greater competence of the immune response to viral infection. The study was inconclusive in determining the role of other viral infections concerning SARS-CoV-2 infection and was designed only on information from the first viral strain. Therefore, it was impossible to understand the course of the disease in the face of (SARS-CoV-2) virus mutations (pathogenic variants). Further studies are needed to understand better the long-term implications of COVID-19 in patients with CF, especially those with severe respiratory compromise and a history of lung transplantation.
[18]	*Journal of Cystic Fibrosis*	5.53	To observe the incidence of SARS-CoV-2 in people with CF in Europe.	Observational study—cohort	One hundred and thirty patients with CF were classified as non-lung transplanted (n = 107) or lung-transplanted (n = 23).	An observational study was conducted based on data collected in 38 countries from the European CF Society Patient Registry. The diagnosis of COVID-19 was based on positive RT-PCR tests for SARS-CoV-2 from 1 February 2020 to 30 June 2020, and follow-up was performed until 7 January 2021. Patients were subdivided according to the need for lung transplantation, and the incidence of infection was calculated by country and age group.	The incidence of COVID-19 was 270/1000 in people with CF. According to this study, the incidence was higher in patients who received lung transplantation than in non-transplanted patients. The incidence was also higher in people with CF than in the general population. Patients with CF had higher hospitalization rates and a greater need for intensive care when compared to the general population. Most individuals with CF progressed to clinical cure (96.2%); however, five patients died (three were lung transplant recipients).	Compared with the general population of the same age, individuals with CF have a higher incidence of SARS-CoV-2 infection and hospitalization, especially those with a history of lung transplantation; thus, this population is at increased risk for death and severe disease outcome. Individuals with CF must continue to protect their health and engage in physical exercise. Health agencies should prioritize vaccination against COVID-19 in this population of patients, regardless of age, including CF, as an increased risk for adverse conditions of SARS-CoV-2 infection.
[27]	*eClinical Medicine*	17.03	To compare the pulmonary function between pre-infection and post-infection periods in SARS-CoV-2 infected patients.	Observational study—cohort	Eighty patients with CF.	A multicenter cohort study of patients with COVID-19 was performed where lung function before and after SARS-CoV-2 infection was compared from March to November 2020.	Most patients were White (n = 70; 87.5%) and never smokers (n = 42; 52.5%). The most patients had mild to moderate COVID-19 disease (n = 60; 75.1%), with 25 (31.2%) requiring hospitalization. There was no difference between the pre-and post-pulmonary function test data. There was also no difference in the pulmonary function test when analyzed by hospitalization and disease severity. After adjusting for potential confounders, interstitial lung disease was independently associated with a decreased forced expiratory volume in the first second, in addition, increasing age and CF were associated with decreasing forced vital capacity when comparing pre-and post-infection pulmonary function tests. Only increasing age was independently associated with a reduction in total lung capacity and diffusing capacity before and after SARS-CoV-2 infection.	Most participants had mild to moderate symptoms of COVID-19; some patients required hospitalization. There was no significant differences in the pulmonary function tests of hospitalized and severely ill patients. There was no exacerbation of SARS-CoV-2 infection in patients with CF; however, there may be a relationship between certain underlying lung diseases and decreased pulmonary function post-infection.
[28]	*Journal of Paediatrics and Child Health*	1.93	To report the experience of SARS-CoV-2 infection outcomes in children with CF.	Letter to the Editor as an observational study	The study comprised 233 children with CF, and only three had a positive diagnosis of COVID-19.	The report provides a one-year overview of the COVID-19 pandemic in children with CF from three pediatric care centers in Oman.	During the COVID-19 pandemic year, only three children from the study population required hospitalization due to COVID-19. Of the children who required hospitalization, only one required supplemental oxygen, non-invasive mechanical ventilation, and intensive care unit monitoring. This patient had moderate–severe CF with forced expiratory volume in the first second of 50% and severe asthma on Omalizumab. This patient required care for two weeks and was treated with oxygen, salbutamol, dual antipseudomonal antibiotics, and chest physiotherapy. The other two children had mild symptoms and remained hospitalized for two to five days.	Three children with CF were hospitalized at the three centers. One child required non-invasive oxygen supplementation and monitoring in an intensive care unit. Children with CF, especially those with poorer lung function, should continue to adhere to COVID-19 containment measures. According to the report, non-urgent bronchoscopy should be postponed in patients with CF due to the development of new variants.
[29]	*Respiratory Medicine*	4.58	To report the clinical outcomes of SARS-CoV-2 infection in a cohort of people with CF and compare these results with a corresponding cohort of people without CF.	Observational study—cohort	The study enrolled 507,388 patients with COVID-19 (422 with CF).	The study described the clinical features of SARS-CoV-2 infection in patients with CF and compared the results with people without CF. The analysis included patients +16 years and RT-PCR positive for SARS-CoV-2.	Only 0.08% (n = 422) of the patients who composed the study had CF. In the CF group, the mean age was 46.6 years, with a predominance of females (n = 225; 53.32%) and Caucasians (White; n = 309; 73.22%). Patients with CF had higher rates of hospitalization, need for intensive care, mechanical ventilation, acute kidney injury, and deaths when compared with patients without CF. Regarding the calculation of data, there was the following propensity for critical admission (RR = 1.56, 95%CI = 1.20–2.04), need for critical care (RR = 1.78, 95%CI = 1.13–2.79), and kidney injury (RR = 1.60, 95%CI = 1.07–2.39) in patients with CF when compared with patients without CF.	Patients with CF are more likely to develop severe disease from SARS-CoV-2 infection; the need for priority vaccination against COVID-19 in this group of patients was described. Ten patients required intensive care. Young patients and a lower prevalence of obesity did not protect patients with CF from serious illness caused by SARS-CoV-2 infection.
[30]	*Boletín Médico del Hospital Infantil de México*	0.71	To report a case of SARS-CoV-2 infection in a child with CF.	Case report	One patient	The case report describes a 22-month-old patient with CF who arrived at the hospital with probable symptoms of COVID-19. The patient received supplemental oxygen therapy, antibiotic therapy, and antiretroviral therapy.	The patient with CF presented tachycardia, polypnea, fever, and oxygen saturation of 87% when he arrived at the hospital. The child displayed breathing difficulties and need for non-invasive ventilation management and high-flow therapy in the intensive care unit. A ventilatory test showed: a peak pressure of 21 cmH_2_O; plateau pressure of 14 cmH_2_O; transpulmonary pressure of seven cmH_2_O; compliance of 0.75; and positive pressure at the end of expiration (autoPEEP) of 0.7. As the patient showed good evolution, extubating was performed 12 h after the admission.	The patient’s evolution was favorable with immediate support and antibiotics in a tertiary care hospital. The study guides on the need to report pediatric cases and the different evolution compared to that presented by adults among patients with CF. Due to the spread of SARS-CoV-2 worldwide, it is essential to share the experience of other methods to improve knowledge about the clinical features, evolution, treatment, and rapid prognosis of those with CF.
[31]	*Journal of Clinical Medicine*	5.41	To describe the clinical expression of COVID-19 in French patients with CF.	Observational study	Patients registered in French CF centers = 7500; and patients with CF and COVID-19 = 31 (post-transplant = 12; and non-transplanted = 19).	An observational and prospective study was carried out with 7500 patients with CF from 47 reference centers in France. The patients were screened for SARS-CoV-2 by RT-PCR through nasopharyngeal swab test, with serological tests only becoming available in May 2020. Data from patients with CF infected by SARS-CoV-2 were collected in a dedicated CF-COVID registry, which is nested within the French CF registry (maintained by the French CF patient association, Vaincre la Mucoviscidose). To account for the potential lack of performance of the RT-PCR for SARS-CoV-2 due to sampling quality or kinetics (sensitivity <60%), a body of evidence was used for assessing COVID-19 diagnosis.	In the period evaluated, 31 patients tested positive for COVID-19, 15 were male with a mean age range of 31 years, and 12 were lung transplant recipients. Most of the patients had CF-related diabetes (n = 19, 61.3%), and a mild lung disease (n = 19, 65.0%), with percent-predicted forced expiratory volume in one second <70%. Only three patients had no symptoms. The most common symptoms for the 28 symptomatic patients were fever (n = 22, 78.6%), fatigue (n = 14, 50.0%), and increased cough (n = 14, 50.0%). Nineteen patients required hospitalization, including eleven lung transplant recipients, seven required oxygen support, and four required intensive care unit hospitalization. Ten patients had severe acute respiratory syndrome. However, all patients presented a good evolution of the disease, and no more serious deleterious effects in the short term were described.	Patients with CF have rarely been diagnosed with COVID-19 in France; however, the rate of underdiagnosis should be evaluated in other studies. The short-term evolution was favorable, with the rare occurrence of the severe acute respiratory syndrome, and death was not described. Post-lung transplant patients need to be followed up with greater attention by the multidisciplinary team due to the tendency to present a more severe clinical course.
[32]	*Plos One*	3.75	To describe the impact of restrictions related to COVID-19 on daily physical activity, airway clearance, and inhalation therapy, questions on COVID-19-compatible symptoms, diagnostic tests, vaccination status, and inquiries about health-related aspects covering the pandemic.	Observational study	Patients with CF = 193 individuals (lung-transplanted = 49; and non-lung transplanted = 144) and patients with CF and COVID-19 = 10 (two lung-transplanted).	A cross-sectional study was carried out during the COVID-19 pandemic in patients with CF in Switzerland. The severity of COVID-19 was classified into stages: I—mild symptoms (dry cough, fatigue, and headache); II—moderate to severe symptoms (dyspnea and hypoxia); III—critical symptoms (discomfort syndrome and heart failure); IV—death.	Most patients were female (53.0%), and only 49 (25.0%) were transplanted. Ten patients with CF reported infection with SARS-CoV-2 being two lung-transplanted. Two patients required hospitalization without the need for invasive mechanical ventilation and death. Forty-six percent reported less physical activity during the pandemic, mostly due to closed fitness facilities (85.0%), lack of motivation (34.0%), and changes in daily structures (21.0%). Among patients with CF without lung transplantation, seven (5.0%) and five (4.0%) reported performing less airway clearance and inhalation therapy during the COVID-19 pandemic, respectively; no death was registered in the study.	The study reveals unfavorable consequences of COVID-19 pandemic restrictions for patients with CF; the long-term effects on physical fitness and lung health are unknown. Strategies to overcome such a situation are necessary.
[33]	*European Respiratory Journal*	33.80	To estimate incidence, describe the clinical presentation, and investigate factors associated with severe outcomes due to SARS-CoV-2 infection in patients with CF.	Observational study	Patients with CF and COVID-19 = 828 (78 with lung transplantation and 750 with no-lung transplantation).	Data from patients with CF from the European CF Society Patient Registry with positive RT-PCR results for SARS-CoV-2 were described. Cases diagnosed by computed tomography, serology, or antigen test without SARS-CoV-2 RT-PCR confirmation were excluded. The clinical features were presented for all patients according to lung transplant status.	Eight hundred and twenty-eight people with CF from 26 countries reported infection with SARS-CoV-2, bringing the incidence to 17.2/1000. The mean age was 24 years, 48.4% were male, and 9.4% were lung transplant recipients. The incidence of SARS-CoV-2 was higher in lung transplant recipients when compared with non-lung transplanted individuals. About seventy-five percent of patients were asymptomatic. Symptomatic infection was associated with age +40 years, at least one p.Phe508del pathogenic variant in the *CFTR* gene, and pancreatic insufficiency. The severity was two to six times greater in transplanted than in non-transplanted patients with CF. Factors predisposing to hospitalization were CF-related diabetes, severe or moderate lung disease, lung transplantation, use of Azithromycin (to treat the *Pseudomonas aeruginosa* infection), and worse lung function.	Clinical symptoms arising from COVID-19 in CF are variable, and pulmonary symptoms resemble those of a common exacerbation in CF. The study described that lung transplantation and moderate to severe previous lung disease were independent risk factors for severe outcomes after SARS-CoV-2 infection. All patients with CF must maintain protective measures to prevent infection with SARS-CoV-2 and must be vaccinated against COVID-19.
[34]	*Journal of Cystic Fibrosis*	5.53	To characterize the SARS-CoV-2 infection in patients with CF.	Observational study	Patients with CF positive for SARS-CoV-2 = 40 (lung-transplanted = 11; and non-transplanted = 29).	Anonymized data were obtained from patients with CF and infected with SARS-CoV-2 between 1 February 2020 and 13 April 2020. Patients with findings suggestive of COVID-19 on computed tomography of the chest and without a positive SARS-CoV-2 RT-PCR were excluded. The following data were collected: sex, age, body mass index, forced expiratory flow in one second, treatment, CF-related complications, microbiology profile, presence of lung transplantation, presence of pregnancy, and characteristics of current health status.	Of the 40 study participants from eight countries, 31 (78.0%) had symptoms of SARS-CoV-2 infection, including fever in 24 (60.0%) patients. A total of 70.0% recovered uneventfully, 30.0% remained unresolved at the notification, and no deaths were reported.	The outcomes of these cases of SARS-CoV-2 in 40 people with CF have been better than initially predicted based on previous respiratory pandemics. However, the medium and long-term impact of COVID-19 is not yet known in patients with CF. Lifetime population mortality may be concerned with the relatively low incidence of SARS-CoV-2 in the CF population, which can be attributed to adherence to social distancing.
[35]	*European Respiratory Journal*	33.80	To evaluate the COVID-19 outcomes in children with underlying chronic diseases.	Observational study	The study enrolled 945 children with COVID-19. Also, 185 children presented with underlying chronic respiratory disease and COVID-19: 63 with asthma; 14 with CF; nine with bronchopulmonary dysplasia; and 99 with other underlying conditions.	Experts created a survey that circulated twice (in April and May 2020) to members of the Pediatric Assembly of the European Respiratory Society (ERS) and through the ERS social media. The research stratified patients by the following conditions: asthma, CF, bronchopulmonary dysplasia, and other respiratory diseases, to evaluate if these diseases are associated with severity due to SARS-CoV-2 infection.	One hundred and seventy-four centers were willing to participate. The SARS-CoV-2 was isolated from 63 (49 with complete information) children with asthma, of whom 16 required no hospitalization and 33 were hospitalized, 19 needed supplemental oxygen, and four required mechanical ventilation. Fourteen (thirteen with complete information) children with CF and COVID-19 were evaluated; ten required no treatment, and four had only minor symptoms. In addition, one patients with CF needed invasive ventilation and two supplemental oxygen. Among the nine children with bronchopulmonary dysplasia and COVID-19, two required no treatment, five required inpatient care and oxygen, and two were admitted to a pediatric intensive care unit requiring invasive ventilation. Data were available from 33 children with other conditions and SARS-CoV-2, of whom 20 required supplemental oxygen and 11 needed non-invasive or invasive mechanical ventilation support.	The low number of children with CF and asthma admitted to the hospital may suggest that these children are not at risk of having COVID-19 with the severe phenotype, which is essential data for guiding governments concerning public health decision-making. Concomitantly, children with other respiratory illnesses and asthma may be at increased risk for disease severity and benefit from being protected against the COVID-19 disease spread.
[36]	*Journal of Cystic Fibrosis*	5.53	To report an asymptomatic case of COVID-19 in an infant with CF.	Letter to the Editor as a case report	One patient	A report was made of a one-month-old child with COVID-19 and CF but asymptomatic. SARS-CoV-2 infection occurred after contact with the infected grandfather, who presented positive SARS-CoV-2 RT-PCR in material collected from the nasopharynx.	The clinical infection course in children was more mild than in adults.	The child progressed asymptomatically to SARS-CoV-2 infection. However, the study concludes that other cases should be evaluated to describe the real impact of COVID-19 among those with CF.
[37]	*Journal of Cystic Fibrosis*	5.53	To report outcomes due to COVID-19 and *Burkholderia cenocepacia* infection in a patient with CF.	Case report	One patient	A report was made of a 34-year-old woman with CF with complications from the disease. The patient sought support from the CF referral center with an elevated temperature and an episode of abnormal 92% oxygen saturation in room air. The patient had a positive RT-PCR for SARS-CoV-2.	The patient needed to move to a sub-intensive care unit. Twelve days after the positive RT-PCR for SARS-CoV-2 from the swab and three days after the disappearance of the symptoms, the RT-PCR tested negative. Thirteen days after admission, the serology for SARS-CoV-2 showed high levels of IgG (13,600 AU/mL) and IgM (5252 AU/mL) antibodies, and the computed tomography of the chest had improved.	Although the SARS-CoV-2 infection had severe manifestation, the patient had a probable outcome with a clinical cure. Infection with SARS-CoV-2 and *B. cenocepacia* warns of the risk of pulmonary complications, and one patient presented a severe course of COVID-19, with signs of interstitial involvement in the examination appointment. Further similar cases can confirm the hypothesis that *B. cenocepacia* colonization should be regarded as a risk factor for severe COVID-19 expression in CF, similar to diabetes mellitus or poor lung function.
[38]	*Respiratory Medicine*	4.58	To describe the impact of SARS-CoV-2 infection in patients with CF in Spain.	Observational study	Eight patients with CF and COVID-19 were enrolled—one lung-transplanted and seven non-lungs transplanted.	A retrospective study including patients with CF was carried out to evaluate the incidence of COVID-19. The research also evaluated the prevention measures against the SARS-CoV-2 infection, the activities of CF reference centers during the COVID-19 pandemic, and the possible reasons for the low incidence of COVID-19 among those with CF.	SARS-CoV-2 infection was detected in eight patients with CF, and one received a lung transplant. The disease incidence was 32/10,000 in patients with CF and 49/10,000 in the general population. No patient died during the study.	Although CF is a risk for the severe development of COVID-19, none of the non-transplanted patients progressed to death, and the incidence of COVID-19 in CF appears to be lower when compared to the general population. Lung transplantation immunosuppression may be an additional risk for mortality in patients with CF. More studies are necessary to evaluate the pandemic progression in the long-term for SARS-CoV-2 infection on CF.
[39]	*Journal of Cystic Fibrosis*	5.53	To measure the SARS-CoV-2 seroprevalence in patients with CF in Belgium.	Observational study—cohort	One hundred forty-nine patients with CF were enrolled—thirty-six were symptomatic, and four had positive serology for COVID-19.	A retrospective study was conducted with patients with CF from 13 April to 19 May 2020. Serum from 149 patients was collected at the CF referral center or sent to a referral center. Patients were contacted by telephone to obtain clinical data about SARS-CoV-2 infection.	Among the patients, only three asymptomatic patients had SARS-CoV-2 virus-related IgG. In one patient hospitalized for COVID-19 (positive molecular test) and another 35 positives, no anti-SARS-CoV-2 antibodies were found.	Although cases with symptomatology were frequent, antibodies were detected in only a minority of patients with CF, reflecting the low incidence of COVID-19 in this population. The study also described the similarity of symptoms of COVID-19 concerning respiratory exacerbations of CF.
[40]	*Journal of Cystic Fibrosis*	5.53	To present data from the first and second waves of SARS-CoV-2 infection in patients with CF.	Letter to the Editor—as an observational study	Patients with CF and positive for SARS-CoV-2 = 87 (first wave = 22; and second wave = 65).	A survey was carried out using an anonymous online form to obtain data from patients with CF and COVID-19 to assess the evolution of the disease. The number of infected individuals, sex, age, the method for diagnosing COVID-19, type of restraint, need for intensive care unit admission, and the number of clinically recovered patients were evaluated in the questionnaire. In the period from February to November 2021, a total of 87 patients with CF were infected with SARS-CoV-2.	Of the 87 patients with CF, only 15 were hospitalized (15/87; 17.2%), and admission to the intensive care unit was required in only two cases (2/87; 2.3%). One patient with CF and COVID-19 died.	The low incidence of COVID-19 in CF may suggest that the disease is a protective factor. The incidence in the first wave was possibly due to the social distancing implemented in families with CF. The percentage of cumulative positivity for SARS-CoV-2 infection in the Italian population with CF was approximately double that observed in the general population. Most patients were asymptomatic or had mild respiratory symptoms.
[41]	*Infection*	7.45	To describe the clinical course and risk factors for severe COVID-19 in patients with CF.	Observational study	Two hundred thirty-six patients with CF and COVID-19.	The Italian Society of CF carried out a prospective study. CF centers collected data and followed up on patients positive (RT-PCR for SARS-CoV-2) for COVID-19 between March and June 2021. Symptoms considered indicative of COVID-19 were: fever, cough, asthenia, anosmia, dysgeusia, shortness of breath, sore throat, headache, myalgia, and diarrhea. The forced expiratory flow was assessed at baseline and the follow-up visit. Subgroup analyses were also performed between symptomatic and asymptomatic patients and those who developed severe and mild COVID-19.	In the study, six patients died, 43 were admitted to the hospital, and four were admitted to the intensive care unit. Pancreatic insufficiency was associated with an increased risk of severe COVID-19 (OR = 4.04, 95%CI = 1.52 to 10.8) as well as forced expiratory volume in one second < 40% (OR = 4.54, 95%CI = 1.56 to 13.2), need of oxygen therapy (OR = 12.3, 95%CI = 2.91 to 51.7), low weight (OR = 2.92, 95%CI = 1.12 to 7.57), organ transplantation (OR = 7.31, 95%CI = 2.59 to 20.7), diabetes mellitus (OR = 2.67, 95%CI = 1.23 to 5.80), and liver disease (OR = 3.67, 95%CI = 1.77 to 7.59), while the use of dornase alfa was associated with a reduced risk (OR = 0.34, 95%CI = 0.13 to 0.88) of severe COVID-19. No significant changes in forced expiratory volume in one second were observed from baseline to a median follow-up of two months.	The study reports the short-term consequence of SARS-CoV-2 infection in CF. Forced expiratory flow to assess the effects of SARS-CoV-2 infection on respiratory function has been found to have limitations, and other more sensitive measures may provide different results, especially in younger patients. In addition, several clinical markers were associated with the severe COVID-19 phenotype.
[42]	*Infection, Disease & Health*	4.05	To report COVID-19 in a complex obstetric patient.	Case report	One patient	Report a case of a 42-year-old transgender patient (genetically female and identified as male) with the forced expiratory flow at 39 weeks of gestation who presented a positive RT-PCR result for SARS-CoV-2.	A 42-year-old transgender patient G2P1 (genetically female, identifying as male) at 39 + 3 weeks gestation with underlying CF (p.Phe508del/p.Arg117His and IVs8 7T/9T polyT variants in the *CFTR* gene) was admitted to a tertiary hospital in Queensland Australia due to possible COVID-19 with symptoms of cough and increased sputum production for five days. His female partner was admitted the day prior due to fever and productive cough following international travel with SARS-CoV-2 detectable on testing.	Patients with CF and pregnancy may have favorable outcomes in the COVID-19 setting. A multidisciplinary team should manage these patients to ensure optimal care, including infection control to prevent transmission and consideration of parental wishes regarding delivery and care of the newborn after birth.
[43]	*Journal of Cystic Fibrosis*	5.53	To report the COVID-19 features in one lung-transplanted patient due to CF.	Case report	One patient	A case report of a 35-year-old man with a history of lung transplantation eight years ago was performed due to the forced expiratory flow decreasing and the presence of fever and dyspnea. A physical examination was performed, pulmonary crackles were noted, and the patient was hospitalized after evaluation by high-resolution computed tomography. On the third day after admission, RT-PCR was positive for SARS-CoV-2. In addition, the patient tested positive for rhinovirus infection.	The SARS-CoV-2 RT-PCR was performed on three successive samples a day apart. Several factors associated with false-negative SARS-CoV-2 RT-PCR results were previously reported. However, nasopharyngeal swabs were unlikely to be optimally performed as two different trained operators sampled them. Furthermore, the hypothesis was raised after the first negative result, and particular attention was taken to the second swabbing. Alternatively, it was possible that the patient was hospitalized for a rhinovirus infection and then acquired a SARS-CoV-2 infection. However, he had been placed in respiratory isolation at admission, and the disease incubation period ranged from two days to 11 days. Furthermore, rhinovirus is rarely associated with severe disease in lung-transplanted patients. His wife presented mild disease, with exclusively pharyngeal symptoms that could be attributed to either rhinovirus or SARS-CoV-2 infection.	The study concludes that lung transplant patients with CF may be at increased risk of developing a severe SARS-CoV-2 infection. Extensive screening is needed in symptomatic patients, including those with milder symptoms. In the context of the COVID-19 pandemic, all these results would help to improve the rapid identification of patients infected with SARS-CoV-2.
[44]	*Indian Journal of Paediatrics*	5.31	To report two cases of children with underlying pulmonary morbidity (CF) who developed severe disease after SARS-CoV-2 infection.	Case report	Two patients with CF	To describe two case reports of children with involvement with a severe course of SARS-CoV-2.	Case 1 (9-year-old boy): Pneumonia with respiratory failure requiring non-invasive ventilatory support. Case 2 (18-month-old boy): Two episodes of severe SARS-CoV-2 lower respiratory infection within two months, requiring high-flow nasal oxygen support.	Children with severe pulmonary involvement may be more likely to develop severe SARS-CoV-2 infection and pulmonary exacerbations. Strict preventive measures must be taken to protect children with CF from COVID-19.
[45]	*Frontiers in Paediatric*	3.56	To analyze the inflammatory cytokines in a patient with CF during the first wave of the COVID-19 pandemic.	Case report	One patient	A clinical case of a 14-year-old girl with CF who had pancreatic insufficiency was described. Also, the study describes the levels of the inflammatory cytokines identified in this patient.	The patient developed chronic lung disease throughout childhood and adolescence with bronchiectasis complicated by *P. aeruginosa.* The tests performed were chest radiography with diffuse thickening of the interlobular septa with parenchymal infiltrates and RT-PCR with a positive result for SARS-CoV-2 infection. The patient with acute-phase COVID-19 had less activation of the immune response, as measured by levels of cytokines or chemokines, such as Interleukin (IL)-6, C-C Motif Chemokine Ligands 2 (CCL2) and 5 (CCL5), and C-X-C Motif Chemokine Ligands 8 (CXCL8), 9 (CXCL9), and 10 (CXCL10) than pediatric and healthy patients referred to the center. Plasma concentrations of IL-6, CCL2, CCL5, CXCL8, and CXCL10 were lower during COVID-19 when compared to levels measured a month earlier. Cytokines remained low after a one-month reassessment, suggesting that the immune response to SARS-CoV-2 was not associated with increased inflammatory response.	The patient with acute-phase COVID-19 had less activation of the immune response, as measured by levels of cytokines or chemokines, than pediatric and healthy patients referred to the center. Cytokines remained low after a one-month reassessment, suggesting that the immune response to SARS-CoV-2 was not associated with increased inflammatory response.
[46]	*BMC Pulmonary Medicine*	3.20	To describe the SARS-CoV-2 incidence in patients with CF for one year.	Observational study	Patients with CF screened for SARS-CoV-2 IgG = 156. Patients with CF positive for SARS-CoV-2 in serology =13.	The profile of anti-SARS-CoV-2 IgG antibodies (serology) and the presence of SARS-CoV-2 in throat saliva or nasopharyngeal swabs of 156 patients with CF were described. Symptoms and other clinical data were recorded for patients with confirmed infection for SARS-CoV-2.	In the study, 13 (8.3%) patients tested positive for SARS-CoV-2 infection, most of them during the second and the beginning of the third wave of the COVID-19 pandemic. Ten patients who tested positive had symptoms linked to COVID-19. The most common symptom was a cough, followed by fatigue and headache. The SARS-CoV-2 infection did not impair lung function. No positive tested patient needed to be hospitalized.	The cumulative frequency of SARS-CoV-2 infection was 8.3% in CF, being lower than in the general population. COVID-19 did not alter the lung disease exacerbation in CF. In addition, delivering data on the immune response in CF will be essential to determine the vaccine priority.
[47]	*Pneumology*	9.21	To evaluate the clinical characteristics and outcomes of SARS-CoV-2 infection in patients with CF managed at home.	Letter to the Editor as a case report	Patients with CF followed at the regional center = 352 (positive diagnosis of COVID-19 = 80).	To describe the features of patients with CF and COVID-19 (positive RT-PCR of nasal/throat material) followed up at the CF center in Florence, Italy, which serves children and adults. Cases were recorded through June 30, 2021. Diagnostic tests were performed when there were symptoms or, in asymptomatic cases, if patients were at risk for positive family contact or work. In the study, pre- and post-infection, the trend of body mass index, or weight/length percentile for infants, and the predicted percentage of forced expiratory volume in one second for patients +6 years, were compared.	Telemedicine consultation took place immediately following the COVID-19 diagnosis and during the active infection (phone calls, monitoring of arterial peripheral oxygen saturation (SpO_2_), and screening for pulmonary symptoms suggestive of COVID-19 were carried out). In the absence of criteria for hospitalization, the authors advised the isolation of the patient and his family, the use of a home pulse oximeter, and, in the case of a reading of less than 92% or respiratory distress signs, the need for hospitalization was suggested. A total of 18 (5.1%) out of 352 patients with CF followed at the regional center suffered from SARS-CoV-2 infection. A total of 13 (72.2%, ten males, mean age at SARS-CoV-2 infection: 27 years, range three months-59 years) out of 18 patients were managed at home. It excluded five patients—two adults who needed hospitalization due to lung transplant and three more cases with persistent fever with SpO_2_ <92%.	In this study, a higher prevalence of SARS-CoV-2 infection in CF than in previous studies was described. However, no cases of infection occurred in the early period of the COVID-19 pandemic, likely due to increased restrictions on social distancing measures. Telemedicine reduced the risk of cross-infection and avoided hospital overcrowding.
[21]	*Clinics*	2.89	To report early SARS-CoV-2 infection after lung transplantation in a patient with CF.	Case report	One patient	A case report of a patient with CF with an indication for lung transplantation was performed. A 21-year-old male patient with no comorbidities was diagnosed with SARS-CoV-2 RT-PCR. The patient was also infected with adenovirus.	The patient showed clinical improvement with supportive treatment without any directed treatment for COVID-19. Although the most probable source of infection was nosocomial, the authors could not determine whether it was from a healthcare professional or the patient’s caregivers, as access to the patient was not restricted when the patient’s condition was worsening. Nosocomial infection must be prevented, especially in countries with many cases of SARS-CoV-2 infection. It is improbable that the donor was the source of the infection because the donor tested negative for SARS-CoV-2 RT-PCR before lung transplantation, an incubation period of two weeks further disfavored this hypothesis in the immunosuppressed patient. In this case, nasopharyngeal swabs of the recipient were not collected before the lung transplantation. Currently, nasopharyngeal swabs are collected for SARS-CoV-2 RT-PCR testing from all patients.	There was a rapid disease evolution in the patient who required mechanical ventilation 72 h after the onset of symptoms. Routine lung transplantation should be maintained with particular attention to donor surveillance and postoperative management. More studies are needed to understand better the role of co-infections with other viruses and the impact of the intensity of immunosuppression on the clinical outcomes of organ recipients with COVID-19.
[50]	*Journal of Cystic Fibrosis*	5.53	To describe the clinical impact of SARS-CoV-2 infection in patients with CF in New York.	Retrospective cohort study	Eight hundred and ten patients with CF were enrolled, and twenty six had a positive diagnosis of COVID-19.	A multicenter retrospective cohort study was conducted to assess the impact of the COVID-19 pandemic on people with CF in the New York metropolitan area from 1 March 2020 to 31 August 2020. The prevalence of COVID-19 was performed by SARS-CoV-2 RT-PCR and IgG antibodies. The following markers were evaluated: clinical characteristics, time in routine outpatient care, and the effect on anxiety and depression in CF.	The prevalence of COVID-19 by SARS-CoV-2 RT-PCR (1.6%) and IgG antibody (12.2%) testing was low. In the study, 58.0% of cases were asymptomatic, and 82.0% were managed at home. Also, 8.0% of the patients were hospitalized and one patient died. In addition, 89.0% of the individuals experienced a delay in care. The prevalence of anxiety increased from 43.0% baseline to 58.0% during the COVID-19 pandemic. In a post hoc analysis, the proportion of patients with diabetes mellitus (38.0% versus 16.0%) and pancreatic insufficiency (96.0% versus 66.0%) were higher in individuals with CF who tested positive for COVID-19. In comparison, CFTR modulator use was lower (46.0% versus 65.0%) in individuals with CF who tested positive for COVID-19.	CF may increase the risk of complications from COVID-19. There was a lower prevalence of COVID-19 when compared to the general population in the assessed region and a low hospitalization rate during the first wave of the pandemic. More studies are needed to understand better the long-term impact of COVID-19 on CF as the general population becomes vaccinated and social contact returns to normal.
[48]	*Case Reports in Women’s Health*	1.48	To report a case of COVID-19 in a pregnant CF carrier with myasthenia gravis.	Case report	One patient	A case report of a 16-year-old Hispanic woman with CF (diagnosed on admission) in her first pregnancy (seven weeks) and with COVID-19 was carried out.	The use of acetylcholinesterase inhibitors, while standard treatment in women with myasthenia gravis and COVID-19 in pregnancy, may have exacerbated mucus production, delaying the ability to extubate the patient from the case report, especially in the context of her CF carrier status.	Regardless of myasthenia gravis, there is a unique opportunity to protect all pregnant women by recommending genetic testing for patients with CF for family planning and identifying a risk factor for severe COVID-19 and mortality.
[49]	*Diagnostics*	3.99	To report the first case of COVID-19 treated with monoclonal anti-spike antibodies in a patient with CF in Romania.	Case report	One patient	A case report of a 17-year-old patient with CF who had not been vaccinated, diagnosed as RT-PCR positive for SARS-CoV-2, was performed.	At the time of evaluation, the patient was afebrile (36.6 °C), with a blood pressure of 121/78 mmHg, heart rate of 85 b.p.m., respiratory rate of 20 cycles/min, and peripheral arterial oxygen saturation of 97% in room air. The electrocardiogram showed no pathological changes. A computed tomography scan of the chest was performed, which revealed isolated central and peripheral ground-glass opacities distributed bilaterally, and was suggestive of mild pneumonia caused by COVID-19.	The study reported the first case of administration of anti-spike SARS-CoV-2 monoclonal antibodies in CF and COVID-19 in the presence of moderate symptoms and good clinical outcomes. As monoclonal antibodies become part of routine clinical practice in Romania and elsewhere, it becomes essential to characterize the profile of patients who are expected to have the most significant benefit and least risk from this type of treatment and prioritize timely diagnosis to allow early therapeutic intervention to prevent progression to severe disease.

%, percentage; <, less than; 95%CI, 95% confidence interval; AU/mL, arbitrary units per milliliter; b.p.m., beats per minute; CFTR, cystic fibrosis transmembrane regulator; cm H_2_O, centimeter of water; G2P1, two pregnancies, one birth; h; hour; IF, impact factor; ref, reference; IgG, immunoglobulin G; IgM, immunoglobulin M; min, minute; mm Hg, millimeter(s) of mercury; N, number of patients; PEEP, positive end-expiratory pressure; RR, relative risk; RT-PCR, real-time polymerase chain reaction; SARS-CoV-2, severe acute respiratory syndrome coronavirus 2.

**Table 2 healthcare-11-01936-t002:** Demographic and clinical characteristics of the hospitalized patients with cystic fibrosis and severe acute respiratory infection (SARI) in Brazil during the coronavirus disease, COVID-19 pandemic.

Markers	Data	N (%)
**Sex**	Male	196 (46.0%)
	Female	230 (54.0%)
**Race**	White	272 (63.8%)
	Black	15 (3.5%)
	Asian	6 (1.4%)
	*Pardos* (Multiracial background)	132 (31.0%)
	Indigenous peoples	1 (0.2%)
**Age ***	≤5 years	96 (22.5%)
	>5 years and ≤10 years	40 (9.4%)
	>10 years and ≤15 years	46 (10.8%)
	>15 years and ≤20 years	36 (8.5%)
	>20 years and ≤25 years	33 (7.7%)
	>25 years and ≤30 years	12 (2.8%)
	>30 years and ≤35 years	13 (3.1%)
	>35 years and ≤40 years	13 (3.1%)
	>40 years and ≤45 years	8 (1.9%)
	>45 years and ≤50 years	10 (2.3%)
	>50 years	119 (27.9%)
**Place of residence**	Urban	385 (90.4%)
	Rural	27 (6.3%)
	Peri-urban	14 (3.3%)
**Nosocomial infection**	Yes	7 (1.6%)
	No	419 (98.4%)
**Clinical symptoms**		
Fever	Yes	259 (60.8%)
Cough	Yes	357 (83.8%)
Sore throat	Yes	33 (7.7%)
Dyspnea	Yes	355 (83.3%)
Respiratory discomfort	Yes	334 (78.4%)
Peripheral arterial oxygen saturation (<95%)	Yes	311 (73.0%)
Diarrhea	Yes	32 (7.5%)
Vomit	Yes	43 (10.1%)
Coryza	Yes	14 (3.3%)
Headache	Yes	21 (4.9%)
Myalgia	Yes	13 (3.1%)
Other clinical symptoms	Yes	58 (13.6%)
**Comorbidities**		
Cardiopathy	Yes	54 (12.7%)
Hematological disease	Yes	2 (0.5%)
Down syndrome	Yes	3 (0.7%)
Liver disease	Yes	9 (2.1%)
Asthma	Yes	22 (5.2%)
Diabetes mellitus	Yes	46 (10.8%)
Neurological disease	Yes	14 (3.3%)
Immunosuppressive disease	Yes	17 (4.0%)
Kidney disease	Yes	10 (2.3%)
Obesity	Yes	7 (1.6%)
**Antiviral to treat flu**	Yes	82 (19.2%)
	No	344 (80.8%)
**Drugs to treat flu—antiviral type**	Oseltamivir	78 (18.3%)
	Zanamivir	1 (0.2%)
	Other	3 (0.7%)
	No	344 (80.8%)
**Intensive care unit**	Yes	135 (31.7%)
	No	291 (68.3%)
**Mechanical ventilatory support**	Invasive	67 (15.7%)
	Non-invasive	244 (57.3%)
	None	115 (27.0%)
**Discharge criterion**	Laboratorial	400 (93.9%)
	Clinical	26 (6.1%)
**Outcome**	Clinical recovery	319 (74.9%)
	Death due to SARI	94 (22.1%)
	Death was not related to SARI	13 (3.1%)
**SARI classification**	SARI due to another viral infection	23 (5.4%)
	SARI due to an unknown etiological agent	286 (67.1%)
	SARI due to SARS-CoV-2 infection	117 (27.5%)

We presented the data using the number of individuals (N) and the percentage (%). <, less than; SARS-CoV-2, severe acute respiratory syndrome coronavirus 2. *, we classified the age according to the Brazilian Cystic Fibrosis Registry.

**Table 3 healthcare-11-01936-t003:** Association between demographical and clinical characteristics of the hospitalized patients with cystic fibrosis according to the severe acute respiratory infection (SARI) groups in Brazil during the coronavirus disease, COVID-19 pandemic.

Markers	Data	SARI Due to Another Viral Infection	SARI Due to an Unknown Etiological Agent	SARI Due to SARS-CoV-2 Infection	Total	*p*
**Sex**	Male	11 (47.8%)	126 (44.1%)	59 (50.4%)	196 (46.0%)	0.489 ^a^
	Female	12 (52.2%)	160 (55.9%)	58 (49.6%)	230 (54.0%)	
**Race**	White	12 (52.2%)	195 (68.2%)	65 (55.6%)	272 (63.8%)	**0.026 ^b^**
	Black	1 (4.3%)	11 (3.8%)	3 (2.6%)	15 (3.5%)	
	Asian	0 (0.0%)	3 (1.0%)	3 (2.6%)	6 (1.4%)	
	*Pardos* (Multiracial background)	9 (39.1%)	77 (26.9%)	46 (39.3%)	132 (31.0%)	
	Indigenous peoples	1 (4.3%)	0 (0.0%)	0 (0.0%)	1 (0.2%)	
**Age ***	≤5 years	16 (69.6%)	70 (24.5%)	10 (8.5%)	96 (22.5%)	**< 0.001 ^b^**
	>5 years and ≤10 years	3 (13.0%)	32 (11.2%)	5 (4.3%)	40 (9.4%)	
	>10 years and ≤15 years	1 (4.3%)	42 (14.7%)	3 (2.6%)	46 (10.8%)	
	>15 years and ≤20 years	2 (8.7%)	26 (9.1%)	8 (6.8%)	36 (8.5%)	
	>20 years and ≤25 years	0 (0.0%)	26 (9.1%)	7 (6.0%)	33 (7.7%)	
	>25 years and ≤30 years	0 (0.0%)	10 (3.5%)	2 (1.7%)	12 (2.8%)	
	>30 years and ≤35 years	0 (0.0%)	11 (3.8%)	2 (1.7%)	13 (3.1%)	
	>35 years and ≤40 years	0 (0.0%)	9 (3.1%)	4 (3.4%)	13 (3.1%)	
	>40 years and ≤45 years	0 (0.0%)	3 (1.0%)	5 (4.3%)	8 (1.9%)	
	>45 years and ≤50 years	0 (0.0%)	2 (0.7%)	8 (6.8%)	10 (2.3%)	
	>50 years	1 (4.3%)	55 (19.2%)	63 (53.8%)	119 (27.9%)	
**Place of residence**	Urban	19 (82.6%)	257 (89.9%)	109 (93.2%)	385 (90.4%)	**0.024 ^b^**
	Rural	1 (4.3%)	18 (6.3%)	8 (6.8%)	27 (6.3%)	
	Peri-urban	3 (13.0%)	11 (3.8%)	0 (0.0%)	14 (3.3%)	
**Nosocomial infection**	Yes	0 (0.0%)	5 (1.7%)	2 (1.7%)	7 (1.6%)	1.000 ^b^
	No	23 (100.0%)	281 (98.3%)	115 (98.3%)	419 (98.4%)	
**Antiviral to treat Flu**	Yes	5 (21.7%)	59 (20.6%)	18 (15.4%)	82 (19.2%)	0.478 ^a^
	No	18 (78.3%)	227 (79.4%)	99 (84.6%)	334 (80.8%)	
**Drugs to treat Flu**	Oseltamivir	5 (21.7%)	55 (19.2%)	18 (15.4%)	78 (18.3%)	0.770 ^b^
	Zanamivir	0 (0.0%)	1 (0.3%)	0 (0.0%)	1 (0.2%)	
	Others	0 (0.00%)	3 (1.0%)	0 (0.0%)	3 (0.7%)	
	No	18 (78.3%)	227 (79.4%)	99 (84.6%)	344 (80.8%)	
**Discharge criterion**	Laboratorial	23 (100.0%)	272 (95.1%)	105 (89.7%)	400 (93.9%)	0.074 ^b^
	Clinical	0 (0.0%)	14 (4.9%)	12 (10.3%)	26 (6.1%)	

We presented the data using the number of individuals (N) and the percentage (%). SARS-CoV-2, severe acute respiratory syndrome coronavirus 2. *, we classified the age according to the Brazilian Cystic Fibrosis Registry. ^a^, we performed the statistical analysis using the chi-square test; ^b^, we performed the statistical analysis using the Fisher’s exact test. We adopted an alpha error of 0.05. We demonstrated the significant *p* using the bold type.

**Table 4 healthcare-11-01936-t004:** Association between the time of symptom onset, time of hospitalization, and time in intensive care unit in the hospitalized patients with cystic fibrosis in Brazil according to the severe acute respiratory infection (SARI) groups and outcomes during the coronavirus disease, COVID-19 pandemic.

SARI Groups	Time between Symptoms Onset and Outcome (Days)	Time between Hospitalization and Outcome (Days)	Time in the Intensive Care Unit (Days)
**SARI due to another viral infection**	14.09 ± 8.71	10.83 ± 8.01	0.08 ± 0.39
**SARI due to an unknown etiological agent**	20.81 ± 23.05	14.53 ± 18.41	1.81 ± 5.42
**SARI due to SARS-CoV-2 infection**	17.92 ± 16.44	11.12 ± 14.36	2.31 ± 4.31
*p*	0.155	**0.007 ^a^**	**<0.001 ^b^**
**Death due to SARI**	20.06 ± 21.90	12.96 ± 20.19	3.86 ± 5.66
**Clinical recovery**	16.39 ± 10.38	9.80 ± 7.05	1.10 ± 2.27
*p* ^c^	0.852	0.549	**<0.001**

We presented the data using the mean and standard deviation. SARS-CoV-2, severe acute respiratory syndrome coronavirus 2. We performed the statistical analysis using the Kruskal–Wallis test and Mann–Whitney test. We adopted an alpha error of 0.05. We demonstrated the significant *p* using the bold type. ^a^, SARI due to an unknown etiological agent was significantly different from SARI due to SARS-CoV-2 infection (*p* = 0.006); ^b^, SARI due to another viral infection was significantly different from SARI due to SARS-CoV-2 infection (*p* < 0.001) and from SARI due to an unknown etiological agent (*p* = 0.015), also, SARI due to an unknown etiological agent was significantly different from SARI due to SARS-CoV-2 infection (*p* = 0.020); ^c^, we evaluated the outcomes for the group of patients infected with the SARS-CoV-2 only.

**Table 5 healthcare-11-01936-t005:** Association between coronavirus disease, COVID-19 vaccination status of the hospitalized patients with cystic fibrosis and severe acute respiratory infection (SARI) groups in Brazil with the risk for death during the coronavirus disease, COVID-19 pandemic.

Data	The Patient Received the COVID-19 Vaccination	The Patient Did Not Receive the COVID-19 Vaccination	Total	*p*
**SARI due to another viral infection group**				
Clinical recovery	1 (6.7%)	14 (93.3%)	15 (100.0%)	*-*
**SARI due to an unknown etiological agent**				
Clinical recovery	25 (32.5%)	52 (67.5%)	77 (100.0%)	0.390 ^a^
Death due to SARI	5 (45.5%)	6 (55.5%)	11 (100.0%)	
Death was not related to SARI	2 (33.3%)	4 (66.6%)	6 (100.0%)	
**SARI due to SARS-CoV-2 infection**				
Clinical recovery	12 (41.4%)	17 (58.6%)	29 (100.0%)	0.139 ^a^
Death due to SARI	14 (70.0%)	6 (30.0%)	20 (100.0%)	
Death was not related to SARI	0 (0.0%)	1 (100.0%)	1 (100.0%)	
**All hospitalized patients with cystic fibrosis**				
Clinical recovery	38 (31.4%)	83 (68.6%)	121 (100.0%)	**0.041 ^a^**
Death due to SARI	19 (61.3%)	12 (38.7%)	31 (100.0%)	
Death was not related to SARI	2 (28.6%)	5 (71.4%)	7 (100.0%)	

We presented the data using the number of individuals (N) and the percentage (%). SARS-CoV-2, severe acute respiratory syndrome coronavirus 2. ^a^, we performed the statistical analysis using the Fisher’s exact test. We adopted an alpha error of 0.05. We demonstrated the significant *p* using the bold type.

**Table 6 healthcare-11-01936-t006:** Association between demographical and clinical characteristics of the hospitalized patients with cystic fibrosis and SARS-CoV-2 infection in Brazil with the risk for death during the coronavirus disease, COVID-19 pandemic.

Markers	Data	Death Due to SARI	Clinical Recovery	Total	*p*
**Sex**	Male	29 (56.9%)	29 (44.6%)	58 (50.0%)	0.262 ^a^
	Female	22 (43.1%)	36 (55.4%)	58 (50.0%)	
**Race**	White	31 (60.8%)	34 (52.3%)	65 (56.0%)	0.819 ^b^
	Black	1 (2.0%)	2 (3.1%)	3 (2.6%)	
	Asian	1 (2.0%)	2 (3.1%)	3 (2.6%)	
	*Pardos* (Multiracial background)	18 (35.3%)	27 (41.5%)	45 (38.8%)	
**Age ***	≤5 years	0 (0.0%)	10 (15.4%)	10 (8.6%)	**< 0.001 ^b^**
	>5 years and ≤10 years	0 (0.0%)	5 (7.7%)	5 (4.3%)	
	>10 years and ≤15 years	1 (2.0%)	2 (3.1%)	3 (2.6%)	
	>15 years and ≤20 years	1 (2.0%)	7 (10.8%)	8 (6.9%)	
	>20 years and ≤25 years	3 (5.9%)	4 (6.2%)	7 (6.0%)	
	>25 years and ≤30 years	2 (3.9%)	0 (0.0%)	2 (1.7%)	
	>30 years and ≤35 years	0 (0.0%)	1 (1.5%)	1 (0.9%)	
	>35 years and ≤40 years	1 (2.0%)	3 (4.6%)	4 (3.4%)	
	>40 years and ≤45 years	1 (2.0%)	4 (6.2%)	5 (4.3%)	
	>45 years and ≤50 years	5 (9.8%)	3 (4.6%)	8 (6.9%)	
	>50 years	37 (72.5%)	26 (40.0%)	63 (54.3%)	
**Place of residence**	Urban	46 (90.2%)	62 (95.4%)	108 (93.1%)	0.297 ^b^
	Rural	5 (9.8%)	3 (4.6%)	8 (6.9%)	
**Nosocomial infection**	Yes	2 (3.9%)	0 (0.0%)	2 (1.7%)	0.191 ^b^
	No	49 (96.1%)	65 (100.0%)	114 (98.3%)	
**Antiviral to treat flu**	Yes	7 (13.7%)	11 (16.9%)	18 (15.5%)	0.797 ^a^
	No	44 (86.3%)	54 (83.1%)	98 (84.5%)	
**Discharge criterion**	Laboratorial	44 (86.3%)	61 (93.8%)	105 (90.5%)	0.209 ^b^
	Clinical	7 (13.7%)	4 (6.2%)	11 (9.5%)	

The patients used only Oseltamivir. We presented the data using the number of individuals (N) and the percentage (%). SARI, severe acute respiratory infection; SARS-CoV-2, severe acute respiratory syndrome coronavirus 2. *, we classified the age according to the Brazilian Cystic Fibrosis Registry. ^a^, we performed the statistical analysis using the chi-square test; ^b^, we performed the statistical analysis using the Fisher’s exact test. We adopted an alpha error of 0.05. We demonstrated the significant *p* using the bold type.

**Table 7 healthcare-11-01936-t007:** Description of the respiratory virus identified in the hospitalized patients with cystic fibrosis according to the severe acute respiratory infection (SARI) groups in Brazil during the coronavirus disease, COVID-19 pandemic.

Result	SARI Due to Viral Infection	SARI Due to an Unknown Etiological Agent	SARI Due to SARS-CoV-2 Infection	Total
Bocavirus	1	0	0	1
Bocavirus + parainfluenza 3 + parainfluenza 4	1	0	0	1
Influenza A	2	0	1 *	3
Parainfluenza 1 + respiratory syncytial virus	1	0	0	1
Parainfluenza 3	2	0	0	2
Parainfluenza 3 + rhinovirus	1	0	0	1
Respiratory syncytial virus	4	0	0	4
Respiratory syncytial virus + rhinovirus	1	0	0	1
Rhinovirus	10	0	2 *	12
SARS-CoV-2	0	0	114	114
RT-PCR without result	0	23	0	23
RT-PCR was negative for viral infection	0	263	0	263
Total	23	286	117	426

We presented the data using the number of individuals (N) and the percentage (%). RT-PCR, real-time polymerase chain reaction; SARS-CoV-2, severe acute respiratory syndrome coronavirus 2. *, the patients also presented a positive result in the RT-PCR for SARS-CoV-2.

## Data Availability

The data used in our manuscript can be obtained at OpenDataSUS (https://opendatasus.saude.gov.br/ (accessed on 23 March 2023)) or on request to the authors.

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
