# Peer review of "Epidemiological Profile of Hospitalized Patients with Cystic Fibrosis in Brazil Due to Severe Acute Respiratory Infection during the COVID-19 Pandemic and a Systematic Review of Worldwide COVID-19 in Those with Cystic Fibrosis"

_healthcare, 2023, doi:10.3390/healthcare11131936_

Round 1

Reviewer 1 Report

The article contains very important information for clinical use. I would like to know if there is information available regarding the mortality and severity of the disease in relation to the first half and the second half of the epidemic period. Is there any information on how many of these CF patients and CF deaths  were vaccinated?

Author Response

Reviewer 1.

Comments and Suggestions for Authors

The article contains very important information for clinical use. I would like to know if there is information available regarding the mortality and severity of the disease in relation to the first half and the second half of the epidemic period. Is there any information on how many of these CF patients and CF deaths were vaccinated?

Reply: Dear reviewer, we added the information in Figure 2 and Table 5. Thanks for providing the suggestion.

Reviewer 2 Report

In this manuscript, Marques and collaborators present and analize two datasets on the epidemiological profile of people with Cystic Fibrosis: (i) the epidemiological profile of people with CF hospitalized in Brazil due to severe acute respiratory infection during the COVID-19 Pandemic; and (ii) a systematic review of COVID-19 prevalence, manifestations, and outcomes in patients with cystic fibrosis. Brazilian data was obtained from a public database and the systematic review was based on publications available on PubMed. Motivations for the study include the lack of characterization of the Brazilian CF population, lack of a comprehensive analysis of the association of CF and COVID-19 and contrasting reports on the severity of SARI outcomes on CF versus general population. The most relevant conclusion was the observation of worse outcomes in CF people exhibiting SARI due to SARS-CoV-2 infection when compared to SARI due to other infections. The authors examine the dataset in significant detail, not only in terms of the variables analyzed, but also in terms of the quantitative and qualitative assessment and framing within the literature.

The work has been carefully designed, executed and reported, providing valuable resources for the CF community, especially in Brazil. I also find this work very timely as the pandemic has now subsided in most of the world. I consider that this work can be accepted for publication after the following minor revisions:

The manuscript does not clearly separate the systematic review of (worldwide) CF population with COVID-19 and the description of hospitalized Brazilian patients. I am afraid the casual reader would consider section 3.1 to refer to Brazilian patients. After all, both the title of the manuscript (“Epidemiological Profile of Hospitalized Patients with Cystic Fibrosis in Brazil…”) and the abstract (which only mentions data from the Brazilian database) are centered in data from Brazil. I don’t want to be mis-interpreted, though: I found the manuscript to be well structured. However, I believe the beginning of sections 3.1 and 3.2 could describe a little bit better what is being presented.

Considering replacing “articles” by “papers” or “publications”.

Replace “Canadian” by “Canada”.

Replace “expansive” by “expensive”.

Author Response

Reviewer 2

Comments and Suggestions for Authors

In this manuscript, Marques and collaborators present and analize two datasets on the epidemiological profile of people with Cystic Fibrosis: (i) the epidemiological profile of people with CF hospitalized in Brazil due to severe acute respiratory infection during the COVID-19 Pandemic; and (ii) a systematic review of COVID-19 prevalence, manifestations, and outcomes in patients with cystic fibrosis. Brazilian data was obtained from a public database and the systematic review was based on publications available on PubMed. Motivations for the study include the lack of characterization of the Brazilian CF population, lack of a comprehensive analysis of the association of CF and COVID-19 and contrasting reports on the severity of SARI outcomes on CF versus general population. The most relevant conclusion was the observation of worse outcomes in CF people exhibiting SARI due to SARS-CoV-2 infection when compared to SARI due to other infections. The authors examine the dataset in significant detail, not only in terms of the variables analyzed, but also in terms of the quantitative and qualitative assessment and framing within the literature.

The work has been carefully designed, executed and reported, providing valuable resources for the CF community, especially in Brazil. I also find this work very timely as the pandemic has now subsided in most of the world. I consider that this work can be accepted for publication after the following minor revisions:

The manuscript does not clearly separate the systematic review of (worldwide) CF population with COVID-19 and the description of hospitalized Brazilian patients. I am afraid the casual reader would consider section 3.1 to refer to Brazilian patients. After all, both the title of the manuscript (“Epidemiological Profile of Hospitalized Patients with Cystic Fibrosis in Brazil…”) and the abstract (which only mentions data from the Brazilian database) are centered in data from Brazil. I don’t want to be mis-interpreted, though: I found the manuscript to be well structured. However, I believe the beginning of sections 3.1 and 3.2 could describe a little bit better what is being presented.

Reply: Dear reviewer, the authors thank you for your important contribution to our study. We added the following minor corrections:

  1. a) we edited the title and included the word “Worldwide” as follows: “Epidemiological profile of hospitalized patients with cystic fibrosis in Brazil due to severe acute respiratory infection during the COVID-19 pandemic and a systematic review of worldwide COVID-19 in those with cystic fibrosis”.
  2. b) we included minor corrections in the abstract. Also, we edited it to comprise the main information about the data obtained in the systematic review which included the patients with cystic fibrosis from the worldwide population.
  3. c) we included as keywords the following words: Systematic Review and Worldwide.
  4. d) we also edited the titles from topics 3.1, 3.2, 3.3, 3.4, and 3.5 as follows:

            3.1. Systematic review of COVID-19 among those with CF – Worldwide population

            3.2. Demographic and clinical characteristics of the Brazilian hospitalized patients with CF and SARI

            3.3. Demographic and clinical characteristics of the Brazilian hospitalized patients with CF according to the SARI group

            3.4. Clinical and demographic characteristics associated with death in Brazilian hospitalized patients with CF and COVID-19

            3.5. Microbiological profile of Brazilian hospitalized patients with CF according to the three groups of SARI.

Considering replacing “articles” by “papers” or “publications”.

Reply: Dear reviewer, the authors corrected the term as recommended.

Replace “Canadian” by “Canada”.

Reply: Dear reviewer, the authors replaced the term “Canadian” with “Canada”.

Replace “expansive” by “expensive”.

Reply: Dear reviewer, the authors replaced the term “expansive” with “expensive”.

Reviewer 3 Report

Dear Authors,

the text is too long, but interesting. It's actually two articles in one: 1) Systematic Review of 4 COVID-19 in those with Cystic Fibrosis; 2) Epidemiological Profile of Hospitalized Patients with Cystic Fibrosis in Brazil due to Severe Acute Respiratory Infection During the COVID-19 Pandemic.

I leave it up to the editors to choose whether to leave the text as is or to encourage authors to split up the content. Certainly, in the latter case, the text must be shortened where is possible, avoiding repetitions, shortening the descriptions in table 1. Graphics in the supplement must be clearly separated; correct citations and cut out half of the "to our best knowledge" statements, which are overused especially in the second part of the text; keyword change, addition.

Author Response

Reviewer 3

Comments and Suggestions for Authors

Dear Authors,

the text is too long, but interesting. It's actually two articles in one: 1) Systematic Review of 4 COVID-19 in those with Cystic Fibrosis; 2) Epidemiological Profile of Hospitalized Patients with Cystic Fibrosis in Brazil due to Severe Acute Respiratory Infection During the COVID-19 Pandemic.

I leave it up to the editors to choose whether to leave the text as is or to encourage authors to split up the content. Certainly, in the latter case, the text must be shortened where is possible, avoiding repetitions, shortening the descriptions in table 1. Graphics in the supplement must be clearly separated; correct citations and cut out half of the "to our best knowledge" statements, which are overused especially in the second part of the text; keyword change, addition.

Reply: Dear reviewer, we performed minor corrections in the text as follows:

a) we excluded minor parts of the text.

b) we performed minor editions in the supplement. However, we did not have more graphics in the supplement.

c) We once used the term “to our best knowledge” in the text.

d) we performed minor corrections in the keywords as recommended by two reviewers.
